# OGT binds a conserved C-terminal domain of TET1 to regulate TET1 activity and function in development

Joel Hrit[1,2], Leeanne Goodrich[1,2], Cheng Li[3,4], Bang-An Wang[5], Ji Nie[6,7,8], Xiaolong Cui[6,7,8], Elizabeth Allene Martin[1,2], Eric Simental[1,2], Jenna Fernandez[9], Monica Yun Liu[10,11], Joseph R Nery[5], Rosa Castanon[5], Rahul M Kohli[10,11], Natalia Tretyakova[9], Chuan He[6,7,8], Joseph R Ecker[5], Mary Goll[3†], Barbara Panning[1]*

[1]Department of Biochemistry and Biophysics, University of California San Francisco, San Francisco, United States; [2]TETRAD Graduate Program, University of California San Francisco, San Francisco, United States; [3]Developmental Biology Program, Memorial Sloan Kettering Cancer Center, New York, United States; [4]Program in Biochemistry and Structural Biology, Cell and Developmental Biology, and Molecular Biology (BCMB Allied program), Weill Cornell Graduate School of Medical Sciences, Cornell University, New York, United States; [5]Genomic Analysis Laboratory and Howard Hughes Medical Institute, Salk Institute for Biological Studies, La Jolla, United States; [6]Department of Chemistry, Howard Hughes Medical Institute, University of Chicago, Chicago, United States; [7]Department of Biochemistry and Molecular Biology, University of Chicago, Chicago, United States; [8]Institute for Biophysical Dynamics, University of Chicago, Chicago, United States; [9]Department of Medicinal Chemistry, University of Minnesota, Minneapolis, United States; [10]Department of Biochemistry and Biophysics, Perelman School of Medicine, University of Pennsylvania, Philadelphia, United States; [11]Department of Medicine, Perelman School of Medicine, University of Pennsylvania, Philadelphia, United States

*For correspondence:
barbara.panning@gmail.com

Present address: †Department of Genetics, University of Georgia, Athens, United States

Competing interests: The authors declare that no competing interests exist.

**Abstract** TET enzymes convert 5-methylcytosine to 5-hydroxymethylcytosine and higher oxidized derivatives. TETs stably associate with and are post-translationally modified by the nutrient-sensing enzyme OGT, suggesting a connection between metabolism and the epigenome. Here, we show for the first time that modification by OGT enhances TET1 activity in vitro. We identify a TET1 domain that is necessary and sufficient for binding to OGT and report a point mutation that disrupts the TET1-OGT interaction. We show that this interaction is necessary for TET1 to rescue hematopoetic stem cell production in tet mutant zebrafish embryos, suggesting that OGT promotes TET1's function during development. Finally, we show that disrupting the TET1-OGT interaction in mouse embryonic stem cells changes the abundance of TET2 and 5-methylcytosine, which is accompanied by alterations in gene expression. These results link metabolism and epigenetic control, which may be relevant to the developmental and disease processes regulated by these two enzymes.
DOI: https://doi.org/10.7554/eLife.34870.001

## Introduction

Methylation at the 5' position of cytosine in DNA is a widespread epigenetic regulator of gene expression. Proper deposition and removal of this mark is indispensable for normal vertebrate development, and misregulation of DNA methylation is a common feature in many diseases (*Guibert and Weber, 2013*; *Smith and Meissner, 2013*). The discovery of the Ten-Eleven Translocation (TET) family of enzymes, which iteratively oxidize 5-methylcytosine (5mC) to 5-hydroxymethylcytosine (5hmC), 5-formylcytosine (5fC), and 5-carboxylcytosine (5caC), has expanded the epigenome (*Tahiliani et al., 2009*; *Ito et al., 2010*; *Kriaucionis and Heintz, 2009*; *He et al., 2011*; *Ito et al., 2011*). These modified cytosines have multiple roles, functioning both as transient intermediates in an active DNA demethylation pathway (*He et al., 2011*; *Guo et al., 2011*; *Cortellino et al., 2011*; *Gao et al., 2013*; *Weber et al., 2016*) and as stable epigenetic marks (*Bachman et al., 2014*; *Bachman et al., 2015*) that may recruit specific readers (*Spruijt et al., 2013*).

One interesting interaction partner of TET proteins is *O*-linked N-acetylglucosamine (*O*-GlcNAc) Transferase (OGT). OGT is the sole enzyme responsible for attaching a GlcNAc sugar to serine, threonine, and cysteine residues of over 1000 nuclear, cytoplasmic, and mitochondrial proteins (*Haltiwanger et al., 1990*; *Hanover et al., 2012*; *Maynard et al., 2016*). Like phosphorylation, *O*-GlcNAcylation is a reversible modification that affects the function of target proteins. OGT's targets regulate gene expression (*Lewis and Hanover, 2014*; *Hardivillé and Hart, 2016*), metabolism (*Hanover et al., 2012*; *Bullen et al., 2014*; *Ruan et al., 2013*), and signaling (*Durning et al., 2016*; *Hanover et al., 2005*), consistent with OGT's role in development and disease (*Hart et al., 2011*; *Levine and Walker, 2016*).

OGT stably interacts with and modifies all three TET proteins and its genome-wide distribution overlaps significantly with TETs (*Vella et al., 2013*; *Deplus et al., 2013*; *Chen et al., 2013*). Two studies in mouse embryonic stem cells (mESCs) have suggested that TET1 and OGT may be intimately linked in regulation of gene expression, as depleting either enzyme reduced the chromatin association of the other and affected expression of its target genes (*Vella et al., 2013*; *Shi et al., 2013*). However, it is unclear to what extent these genome-wide changes are direct effects of perturbing the TET1-OGT interaction. Further work is necessary to uncover the biological importance of the partnership between TET1 and OGT.

In this work, we map the interaction between TET1 and OGT to a small C-terminal region of TET1, which is both necessary and sufficient to bind OGT. We show for the first time that OGT modifies the catalytic domain of TET1 in vitro and enhances its catalytic activity. We also use mutant TET1 to show that the TET1-OGT interaction promotes TET1 function in the developing zebrafish embryo. Finally, we show that in mESCs a mutation in TET1 that impairs its interaction with OGT results in alterations in gene expression and in abundance of 5mC and TET2. Together these results suggest that OGT regulates TET1 activity, indicating that the TET1-OGT interaction may be two-fold in function – allowing TET1 to recruit OGT to specific genomic loci and allowing OGT to modulate TET1 activity.

## Results

### A short C-terminal region of TET1 is necessary for binding to OGT

TET1 and OGT interact with each other and are mutually dependent for their localization to chromatin(*Vella et al., 2013*). To understand the role of this association, it is necessary to specifically disrupt the TET1-OGT interaction. All three TETs interact with OGT via their catalytic domains (*Deplus et al., 2013*; *Chen et al., 2013*; *Ito et al., 2014*). We sought to identify the region within the TET1 catalytic domain (TET1 CD) responsible for binding to OGT. The TET1 CD consists of a cysteine-rich N-terminal region necessary for co-factor and substrate binding, a catalytic fold consisting of two lobes separated by a spacer of unknown function, and a short C-terminal region also of unknown function (*Figure 1A*). We transiently transfected HEK293T cells with FLAG-tagged mouse TET1 CD constructs bearing deletions of each of these regions, some of which failed to express (*Figure 1B*). Because HEK293T cells have low levels of endogenous OGT, we also co-expressed His-tagged human OGT (identical to mouse at 1042 of 1046 residues). TET1 constructs were immunoprecipitated (IPed) using a FLAG antibody and analyzed for interaction with OGT. We found that deletion of only the 45 residue C-terminus of TET1 (hereafter C45) prevented detectable interaction

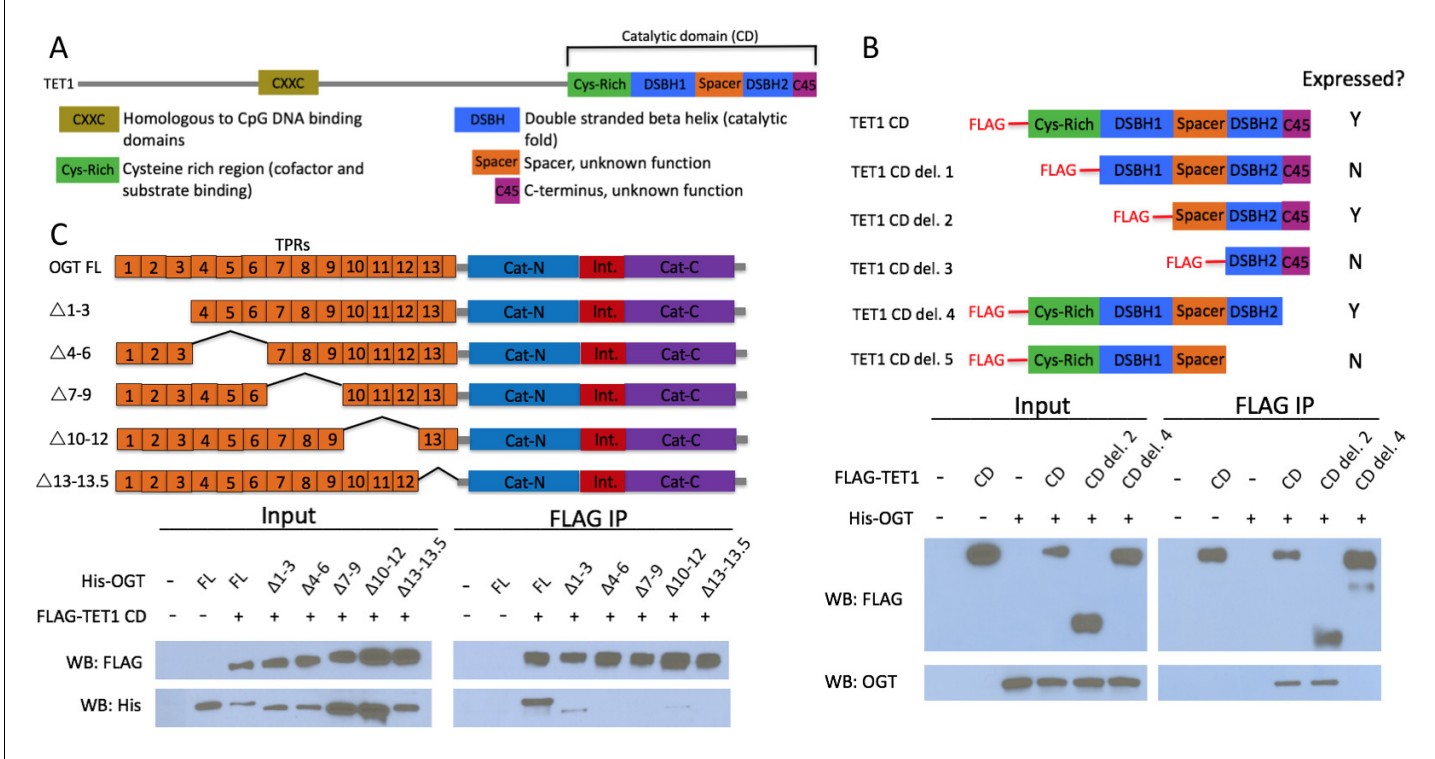

**Figure 1.** The short TET1 C-terminus is required for interaction with OGT. (**A**) Domain architecture of TET1. (**B**) Diagram of FLAG-tagged TET1 CD constructs expressed in HEK293T cells (upper). FLAG and OGT western blot of inputs and FLAG IPs from HEK293T cells transiently expressing FLAG-TET1 CD truncations and His-OGT (lower). (**C**) Diagram of His-tagged OGT constructs expressed in HEK293T cells (upper). FLAG and His western blot of input and FLAG IPs from HEK293T cells transiently expressing FLAG-TET1 CD and His-OGT TPR deletions (lower).

DOI: https://doi.org/10.7554/eLife.34870.002

The following figure supplement is available for figure 1:

**Figure supplement 1.** TET1 C45 is necessary for interaction with endogenous OGT FLAG and OGT western blot of inputs and FLAG IPs from HEK293T cells transiently expressing FLAG-TET1 CD or FLAG-TET1 CD ΔC45 (diagrammed in the upper panel).

DOI: https://doi.org/10.7554/eLife.34870.003

with OGT (*Figure 1B*, TET1 CD del. 4). To exclude the possibility that this result is an artifact of OGT overexpression, we repeated the experiment overexpressing only TET1. TET1 CD, but not TET1 CD lacking C45, interacted with endogenous OGT, confirming that the C45 is necessary for this interaction (*Figure 1—figure supplement 1*).

OGT has two major domains: the N-terminus consists of 13.5 tetratricopeptide repeat (TPR) protein–protein interaction domains, and the C-terminus contains the bilobed catalytic domain (*Figure 1C*). We made internal deletions of several sets of TPRs to ask which are responsible for binding to the TET1 CD. We co-transfected HEK293T cells with FLAG-TET1 CD and His6-tagged OGT constructs and performed FLAG IP and western blot as above. We found that all the TPR deletions tested impaired the interaction with TET1 CD, with deletion of TPRs 7 – 9, 10 – 12, or 13 – 13.5 being most severe (*Figure 1C*). This result suggests that all of OGT's TPRs may be involved in binding to the TET1 CD, or that deletion of a set of TPRs disrupts the overall structure of the repeats in a way that disfavors binding.

## Conserved residues in the TET1 C45 are necessary for the TET1-OGT interaction

An alignment of the TET1 C45 region with the C-termini of TET2 and TET3 revealed several conserved residues (*Figure 2A*). We mutated clusters of three conserved residues in the TET1 C45 of FLAG-tagged TET1 CD (*Figure 2B*) and co-expressed these constructs with His-OGT in HEK293T cells. FLAG pulldowns revealed that two sets of point mutations disrupted the interaction with OGT:

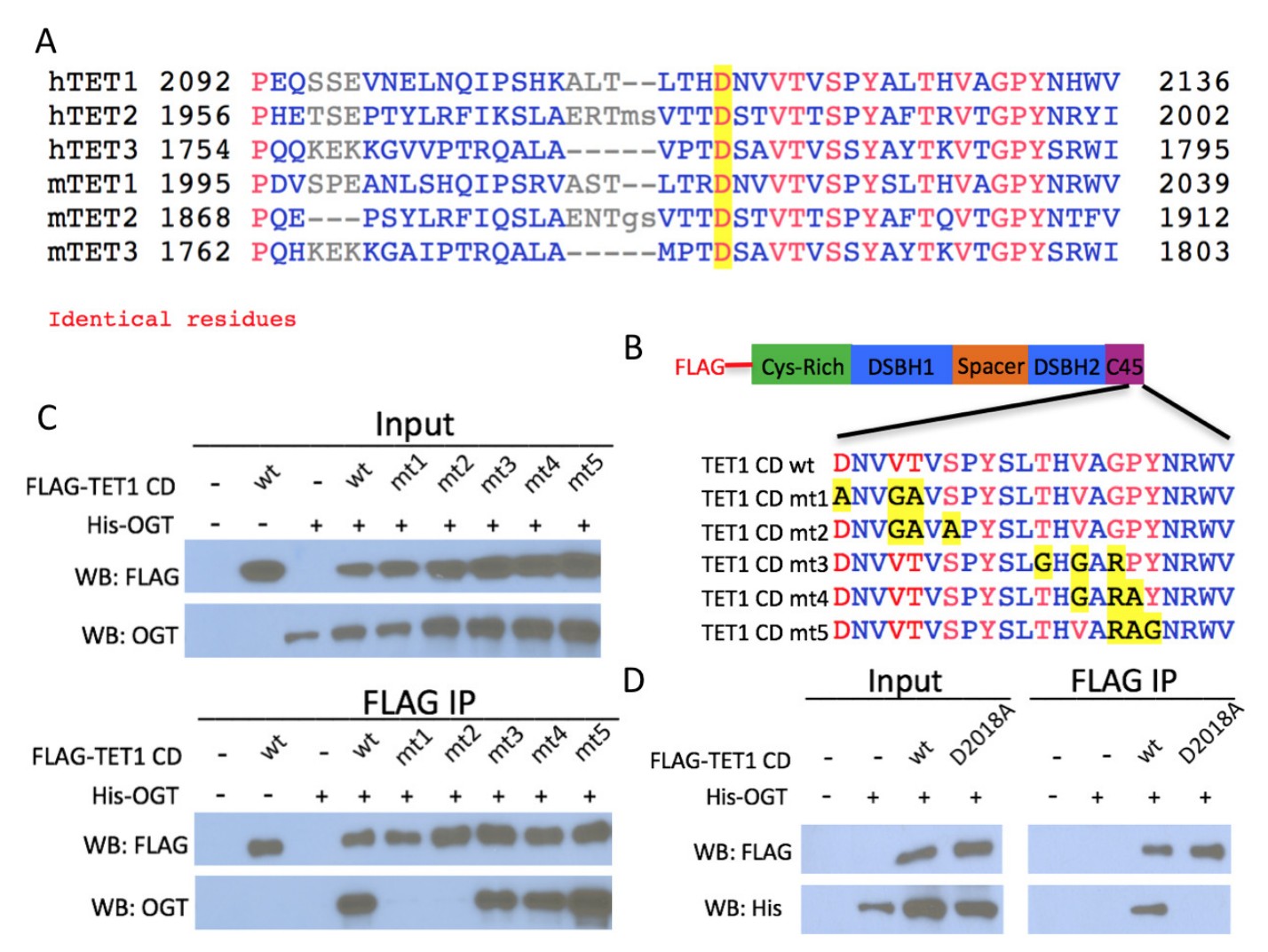

**Figure 2.** Conserved residues in the TET1 C45 are necessary for the TET1-OGT interaction. (**A**) Alignment of the C-termini of human (h) and mouse (m) TETs 1, 2, and 3. A conserved aspartate residue mutated in D is highlighted. (**B**) Diagram of FLAG-tagged TET1 CD constructs expressed in HEK293T cells. (**C**) FLAG and OGT western blot of inputs and FLAG IPs from HEK293T cells transiently expressing FLAG-TET1 CD triple point mutants and His-OGT. (**D**) FLAG and His western blot of inputs and FLAG IPs from HEK293T cells transiently expressing His-OGT and FLAG-TET1 CD or FLAG-TET1 CD D2018A.

DOI: https://doi.org/10.7554/eLife.34870.004

mutation of D2018, V2021, and T2022, or mutation of V2021, T2022, and S2024 (*Figure 2C*, mt1 and mt2). These results suggested that the residues between D2018 and S2024 are crucial for the interaction between TET1 and OGT. Further mutational analysis revealed that altering D2018 to A (D2018A) eliminated detectable interaction between FLAG-tagged TET1 CD and His-OGT (*Figure 2D*).

## The TET1 C-terminus is sufficient for binding to OGT

Having shown that the TET1 C45 is necessary for the interaction with OGT, we next examined if it is also sufficient to bind OGT. We fused the TET1 C45 to the C-terminus of GFP (*Figure 3A*) and investigated its interaction with OGT. We transiently transfected GFP or GFP-C45 into HEK293T cells and pulled down with a GFP antibody. We found that GFP-C45, but not GFP alone, bound OGT (*Figure 3B*), indicating that the TET1 C45 is sufficient for interaction with OGT.

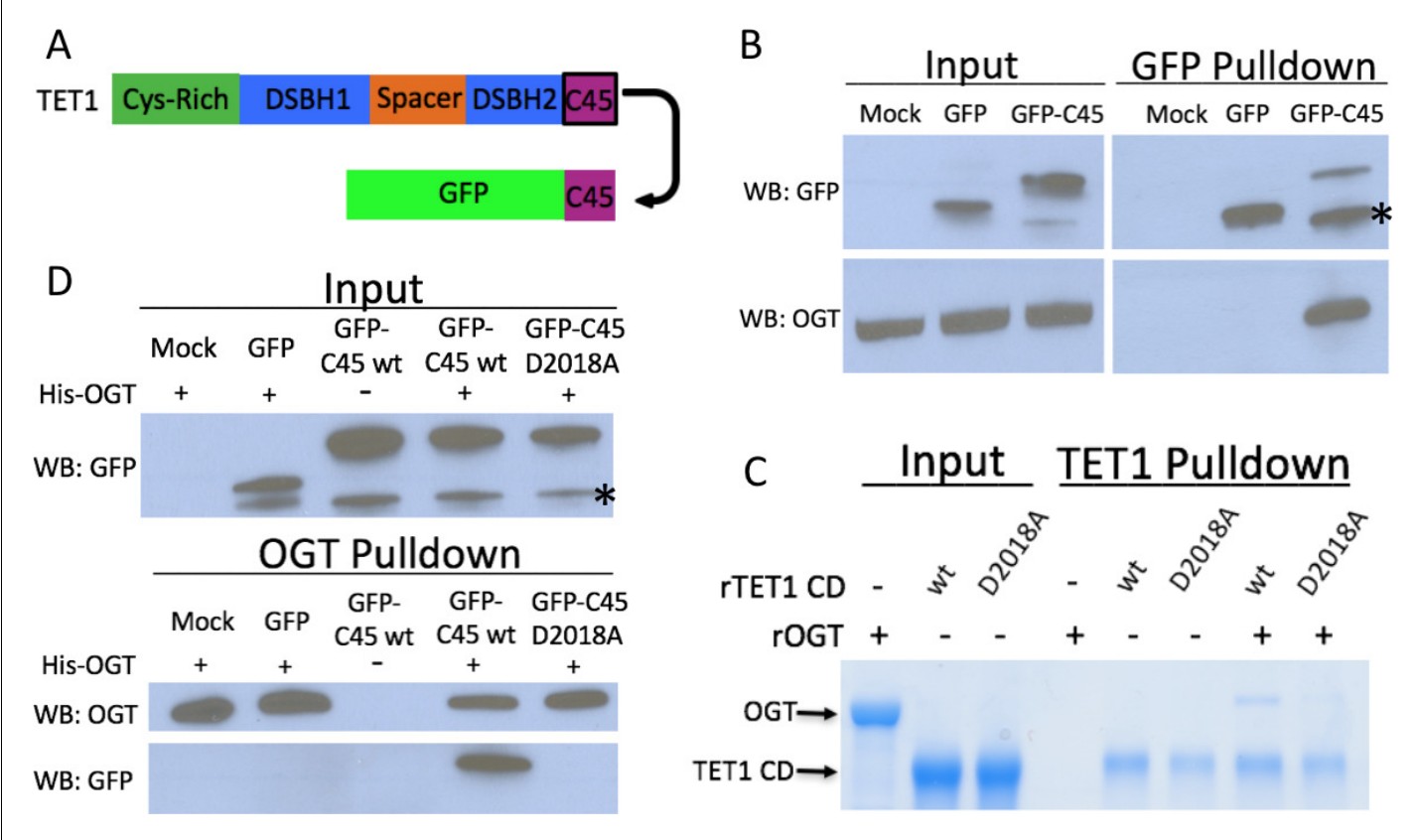

**Figure 3.** The TET1 C45 is sufficient for interaction with OGT in cells and in vitro. (A) Schematic of the TET1 C45 fusion to the C-terminus of GFP. (B) GFP and OGT western blot of inputs and GFP IPs from HEK293T cells transiently expressing GFP or GFP-TET1 C45. *Truncated GFP. (C) Coomassie stained protein gel of inputs and TET1 IPs from in vitro binding reactions containing rOGT and rTET1 CD wild type or D2018A. No UDP-GlcNAc was included in these reactions. (D) GFP and OGT western blot of inputs and OGT IPs from in vitro binding reactions containing rOGT and in vitro translated GFP constructs. *Truncated GFP.
DOI: https://doi.org/10.7554/eLife.34870.005

To determine if the interaction between TET1 CD and OGT is direct, we employed recombinant proteins in pulldown assays using beads conjugated to a TET1 antibody. We used recombinant human OGT (rOGT) isolated from *E. coli* and recombinant mouse TET1 catalytic domain (aa1367-2039), either wild type (rTET1 wt) or D2018A (rD2018A) purified from sf9 cells. rTET1 wt, but not beads alone, pulled down rOGT, indicating a direct interaction between these proteins (*Figure 3C*). rD2018A did not pull down rOGT, consistent with our mutational analysis in cells. Then we used an in vitro transcription/translation extract to produce GFP and GFP-C45, incubated each with rOGT, and found that the TET1 C45 is sufficient to confer binding to rOGT (*Figure 3D*). The D2018A mutation in the GFP-C45 was also sufficient to prevent rOGT binding (*Figure 3D*), consistent with the behavior of TET1 CD D2018A in cells. Together these results indicate that the TET1-OGT interaction is direct and mediated by the TET1 C45.

## The D2018A mutation impairs TET1 CD stimulation by OGT

We employed the D2018A mutation to investigate the effects of perturbing the TET1-OGT interaction on rTET1 activity. rTET1 wt and rD2018A catalyzed formation of 5hmC on an in vitro methylated lambda DNA substrate (*Figure 4A*). Incubation with rOGT and OGT's cofactor UDP-GlcNAc resulted in *O*-GlcNAcylation of rTET1 wt but not rD2018A (*Figure 4B*).

To explore whether *O*-GlcNAcylation affects TET1 CD activity, we incubated rTET1 wt and rD2018A with UDP-GlcNAc and rOGT individually or together and assessed 5hmC production (*Figure 4C–F*, *Figure 4—source data 1*). Addition of UDP-GlcNAc did not significantly affect activity

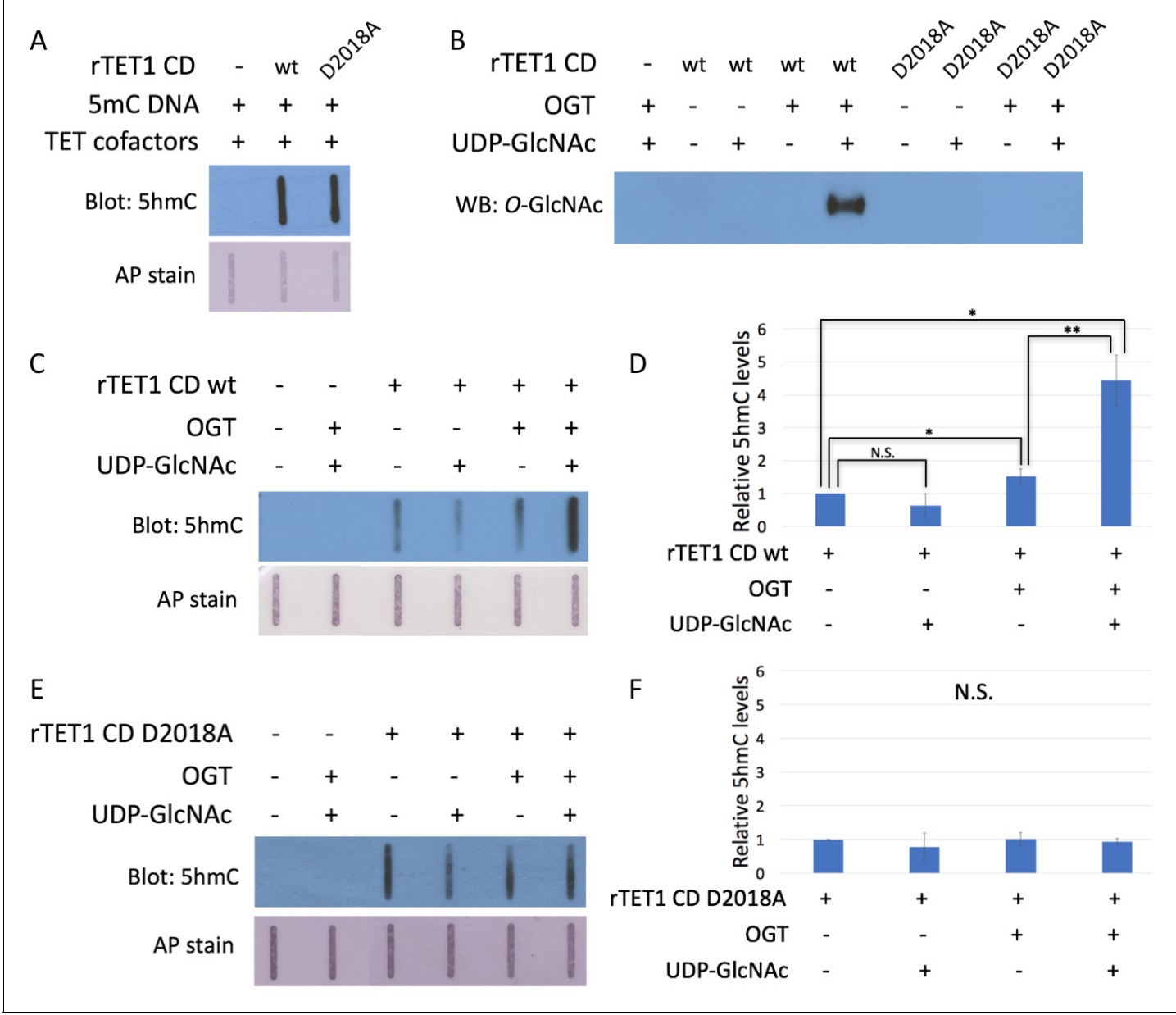

**Figure 4.** The D2018A mutation impairs TET1 CD stimulation by OGT. (A) 5hmC slot blot of biotinylated 5mC containing lambda DNA from rTET1 CD activity assays. Alkaline phosphatase staining was used to detect biotin as a loading control. (B) Western blot for *O*-GlcNAc in in vitro O-GlcNAcylation reactions. (C) 5hmC slot blot of biotinylated 5mC containing lambda DNA from rTET1 wt activity assays. Alkaline phosphatase staining was used to detect biotin as a loading control. (D) Quantification of 5hmC levels from rTET1 wt activity assays. Results are from 3 to 5 slot blots and normalized to rTET1 wt alone. (E) 5hmC slot blot of biotinylated 5mC containing lambda DNA from rD2018A activity assays. Alkaline phosphatase staining was used to detect biotin as a loading control. (F) Quantification of 5hmC levels from rD2018A activity assays. Results are from 3 to 5 slot blots and normalized to rD2018A alone. Error bars denote s.d. *p<0.01, **p<0.01, N.S. – not significant.

DOI: https://doi.org/10.7554/eLife.34870.006

The following source data is available for figure 4:

**Source data 1.** For *Figure 4D* and 4F Quantification of 5hmC levels from activity assays with rTET1 wt.
DOI: https://doi.org/10.7554/eLife.34870.007

of rTET1 wt or rD2018A. Incubation with rOGT alone slightly enhanced 5hmC synthesis by rTET1 wt (1.3 to 1.7-fold), but not rD2018A. We observed robust stimulation (4 to 5-fold) when rTET1 wt but not rD2018A was incubated with rOGT and UDP-GlcNAc. These results suggest that while the TET1-OGT protein-protein interaction may slightly enhance TET1's activity, the *O*-GlcNAc modification is responsible for the majority of the observed stimulation.

## The TET-OGT interaction promotes TET1 function in the zebrafish embryo

We used zebrafish as a model system to ask whether the D2018A mutation affects TET function during development. Deletion analysis of *tet*s in zebrafish showed that Tet2 and Tet3 are the most important in development, while Tet1 contribution is relatively limited (*Li et al., 2015*). Deletion of both *tet2* and *tet3* (*tet2/3* DM) causes a severe decrease in 5hmC levels accompanied by larval lethality owing to abnormalities including defects in hematopoietic stem cell (HSC) production. Reduced HSC production is visualized by reduction in the transcription factor *runx1*, which marks HSCs in the dorsal aorta of wild-type embryos, but is largely absent from this region in *tet2/3* DM embryos. 5hmC levels and *runx1* expression are rescued by injection of human TET2 or TET3 mRNA into one-cell-stage embryos (*Li et al., 2015*).

Given strong sequence conservation among vertebrate TET/Tet proteins, we asked if over expression of mouse Tet1 mRNA could also rescue HSC production in *tet2/3* DM zebrafish embryos and if this rescue is OGT interaction-dependent. To this end, *tet2/3* DM embryos were injected with wild type or D2018A mutant encoding mouse Tet1 mRNA at the one cell stage. At 30 hr post fertilization (hpf) embryos were fixed and the presence of *runx1* positive HSCs in the dorsal aorta was assessed by in situ hybridization (*Figure 5A*). Tet1 wild type mRNA significantly increased the percentage of embryos with strong *runx1* labeling in the dorsal aorta (high *runx1*), while Tet1 D2018A mRNA failed to rescue *runx1* positive cells (*Figure 5A–B*, *Figure 5—source data 1*). We also performed dot blots with genomic DNA from these embryos to measure levels of 5hmC (*Figure 5C*). On average, embryos injected with wild type Tet1 mRNA showed a modest but significant increase in 5hmC relative to uninjected *tet2/3* DM embryos, while injection of TET1 D2018A mRNA did not show a significant increase (*Figure 5D*). These results suggest that the TET1-OGT interaction promotes both TET1's catalytic activity and its ability to rescue *runx1* expression in this system.

## The D2018A mutation alters gene expression and 5mC levels in mESCs

Given the defect of TET1 D2018A in the zebrafish system, we decided to explore the effect of this mutation in mammalian cells. To this end, we generated a D2018A mutation in both copies of the *Tet1* gene (*Figure 6A*) in mESCs (*Figure 6—figure supplement 1*). A FLAG tag was also introduced onto the C-terminus of wild type (WT) or D2018A mutant (D2018A) TET1. We first tested whether D2018 was necessary for the TET1-OGT interaction in the context of endogenous full length TET1 in these cells. FLAG pulldowns revealed that the D2018A mutation reduced, but did not eliminate, co-IP of OGT with TET1 (*Figure 6B*). Levels of TET2 protein were significantly increased in D2018A cells compared to WT (*Figure 6C*), suggesting the cells may be compensating for impaired TET1 function by producing more TET2.

To determine whether the TET1 D2018A mutation affected gene expression, we compared WT and D2018A mESCs using RNA-seq. We identified 378 genes whose expression changed by 2-fold or more in D2018A cells compared to WT (157 upregulated and 221 downregulated)(*Figure 6D*, *Supplementary file 1*). In spite of the increased TET2 protein levels in D2018A cells, we did not observe increased abundance of *Tet2* transcripts (*Figure 6E*, *Figure 6—source data 1*), nor increased stability of TET2 protein in D2018A cells compared to WT (*Figure 6—figure supplement 2*, *Figure 6—figure supplement 2—source data 1*).

To examine how the TET1 D2018A mutation affected DNA modifications, we used LC-MS/MS to measure levels of 5mC and 5hmC in WT and D2018A mESCs. The total amount of 5mC was about 25% lower in D2018A cells compared to WT, while levels of 5hmC were not significantly different (*Figure 6F*, *Figure 6—source data 2*).

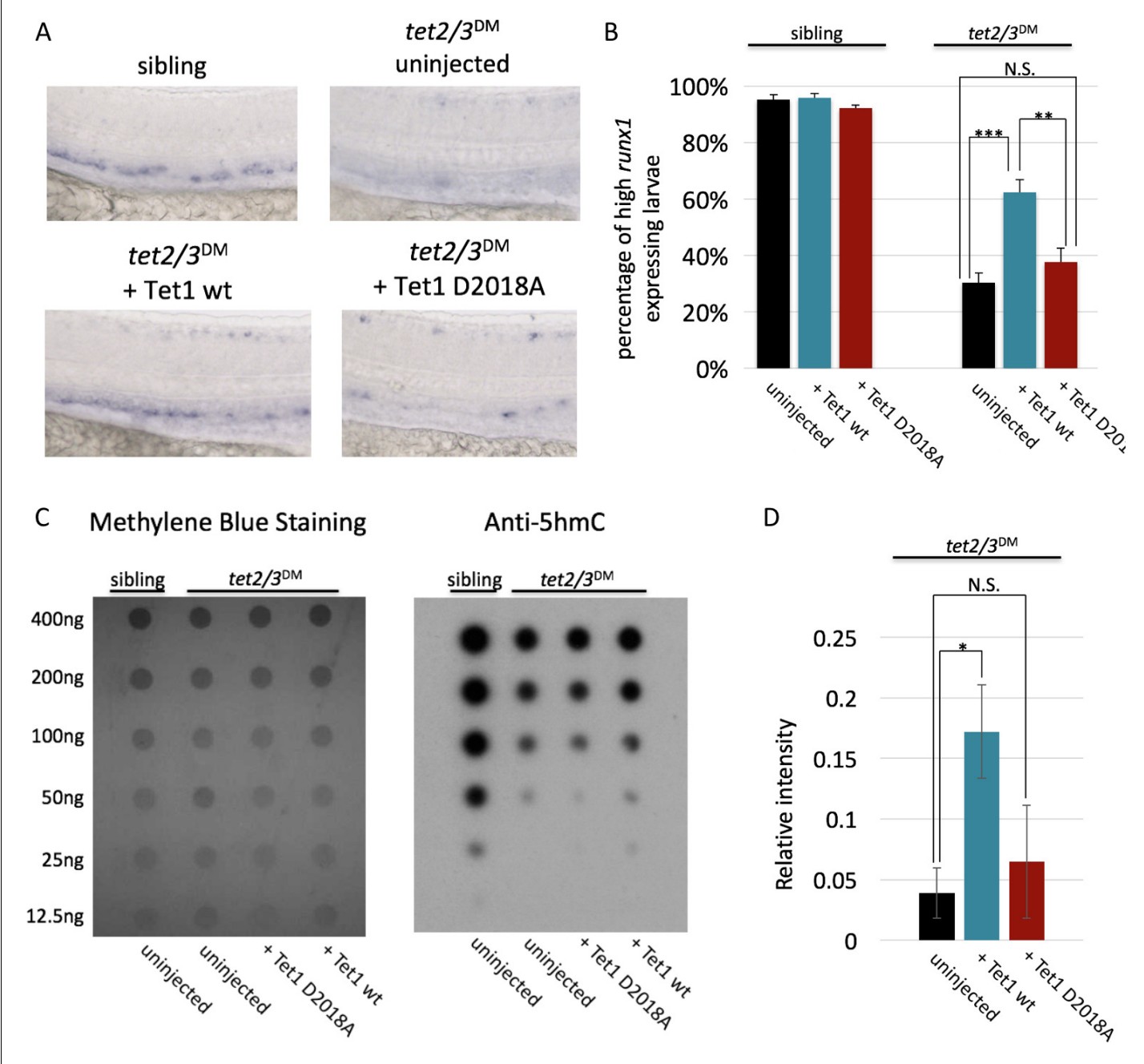

**Figure 5.** The TET1-OGT interaction promotes TET1 function in the zebrafish embryo. (A) Representative images of *runx1* labeling in the dorsal aorta of wild type or *tet2/3*DM zebrafish embryos, uninjected or injected with mRNA encoding mouse Tet1 wild type or D2018A. (B) Percentage of embryos with high *runx1* expression along the dorsal aorta. (C) 5hmC dot blot of genomic DNA from wild type or *tet2/3*DM zebrafish embryos injected with Tet1 wild type or D2018A mRNA. Methylene blue was used as a loading control. (D) Quantification of 5hmC levels from three dot blots, normalized to methylene blue staining. Error bars denote s.d. *$p < 0.05$, **$p < 0.01$, ***$p < 0.001$, N.S. – not significant.

DOI: https://doi.org/10.7554/eLife.34870.008

The following source data is available for figure 5:

**Source data 1.** For *Figure 5B* Quantification of *runx1* levels in zebrafish embryos used to generate graph in *Figure 5B*.
DOI: https://doi.org/10.7554/eLife.34870.009

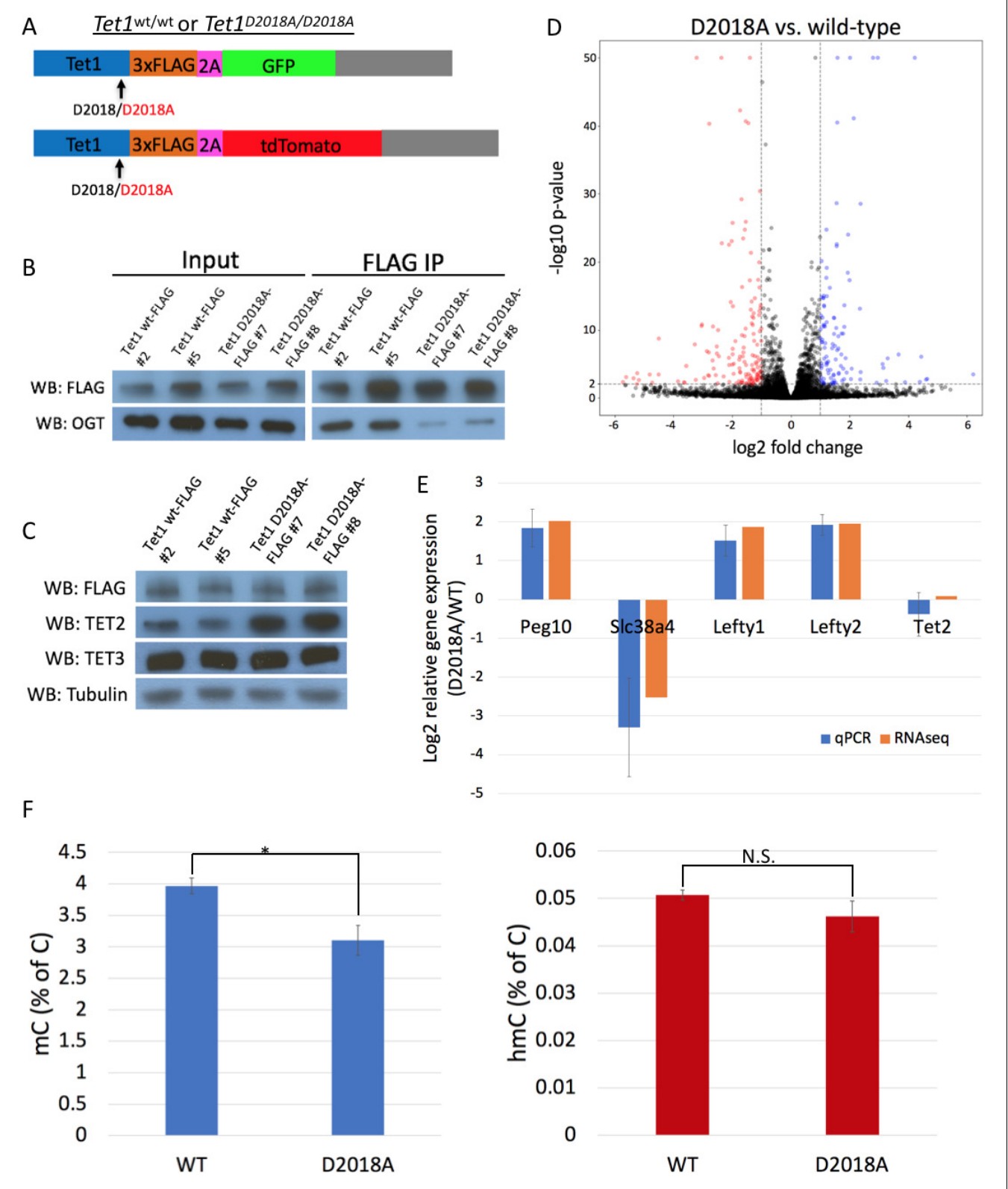

**Figure 6.** The D2018A mutation alters gene expression and 5mC levels in mESCs. (**A**) Schematic of WT and D2018A mESC lines. (**B**) FLAG and OGT western blot of inputs and FLAG IPs from WT and D2018A mESCs. (**C**) Western blots for FLAG, TET2, and TET3 of protein extracts from WT and D2018A mESCs. (**D**) Volcano plot showing differentially expressed genes in D2018A vs. WT mESCs. Red: decreased expression (log2 fold change > −1, Benjamini-Hochberg adjusted p-value<0.01). Blue: increased expression (log2 fold change >1, Benjamini-Hochberg adjusted p-value<0.01) E) qPCR

*Figure 6 continued on next page*

*Figure 6 continued*

analysis of selected differentially expressed genes. (F) Mass spec quantification of mC and hmC levels in WT and D2018A cells. Error bars denote s.d. *p<0.05, N.S. – not significant.

DOI: https://doi.org/10.7554/eLife.34870.010

The following source data and figure supplements are available for figure 6:

**Source data 1.** For *Figure 6E* Quantification of gene expression by RT-qPCR used for graph in *Figure 6E*.

DOI: https://doi.org/10.7554/eLife.34870.014

**Source data 2.** For *Figure 6F* Mass spectrometry quantification of 5mC and 5hmC in WT and D2018A mESCs, used to generate graph in *Figure 6F*.

DOI: https://doi.org/10.7554/eLife.34870.015

**Figure supplement 1.** Generation of mESC lines.

DOI: https://doi.org/10.7554/eLife.34870.011

**Figure supplement 2.** Analysis of TET2 protein stability.

DOI: https://doi.org/10.7554/eLife.34870.012

**Figure supplement 2—source data 1.** For *Figure 6—figure supplement 2* Quantification of western blots for TET2 and Tubulin from WT and D2018A mESCs treated with cycloheximide, used to generate graph in *Figure 6—figure supplement 2*.

DOI: https://doi.org/10.7554/eLife.34870.016

**Figure supplement 3.** Analysis of 25 kb deletion in WT cells Coverage from WGBS.

DOI: https://doi.org/10.7554/eLife.34870.013

**Figure supplement 3—source data 1.** For *Figure 6—figure supplement 3C* Quantification of gene expression by RT-qPCR used for graph in *Figure 6—figure supplement 3C*.

DOI: https://doi.org/10.7554/eLife.34870.017

**Figure supplement 3—source data 2.** For *Figure 6—figure supplement 3D* Mass spectrometry quantification of 5mC in mESCs, used to generate graph in *Figure 6—figure supplement 3D*.

DOI: https://doi.org/10.7554/eLife.34870.018

## The D2018A mutation redistributes 5hmC and reduces 5mC levels

To determine if perturbing the TET1-OGT interaction affected the distribution of CpG modifications, we performed 5hmC-Seal and whole genome bisulfite sequencing (WGBS) on WT and D2018A mESCs. Our 5hmC-Seal analysis detected very few hydroxymethylated peaks, consistent with the low levels of 5hmC detected by mass spec (*Figure 6F*). We identified 76 differentially hydroxymethylated regions (DhMRs), which were enriched in genic regions over intergenic regions (*Figure 7A*). 95 genes were associated with a change in 5hmC levels (DhMGs) – 46 with more 5hmC in D2018A mESCs and 49 with less 5hmC (*Figure 7B*). Metagene analysis showed no dramatic changes in 5hmC distribution around genes (*Figure 7C*). Overall, this data suggests a redistribution of 5hmC in the D2018A mESCs at a small subset of genes. The small number of 5hmC changes precludes a statistically meaningful comparison with the differentially expressed genes (DEGs) identified by RNA-seq.

Consistent with our mass spec results, WGBS showed reduced levels of 5mC + 5 hmC in D2018A cells compared to WT (*Figure 8A,B*). We found no evidence of a change in the distribution of CpG modifications; rather, 5mC + 5 hmC was reduced genome-wide in D2018A cells (*Figure 8C*). To examine whether promoter methylation differences correlate with differential gene expression, we compared the difference in average CpG modification at each promoter to the fold change of the corresponding gene. We found a very small, but statistically significant, negative correlation between promoter CpG modification and gene expression (Pearson r = −0.02, p=0.025)(*Figure 8D*, *Figure 8—figure supplement 1*).

The WGBS analysis identified 42,725 differentially methylated regions (DMRs)(*Figure 8E–G*), none of which overlapped with the 76 DhMRs. Similar to DhMRs, DMRs were significantly enriched in exons over other genomic regions (*Figure 8E*). Next we asked if there was a correlation between gene body DMRs and gene expression. We identified 3587 genes that contained two or more DMRs between the transcription start and end sites. This group was enriched for DEGs (*Figure 8H*), suggesting that changes in gene body cytosine modifications may underlie some of the gene expression differences between WT and D2018A mESCs.

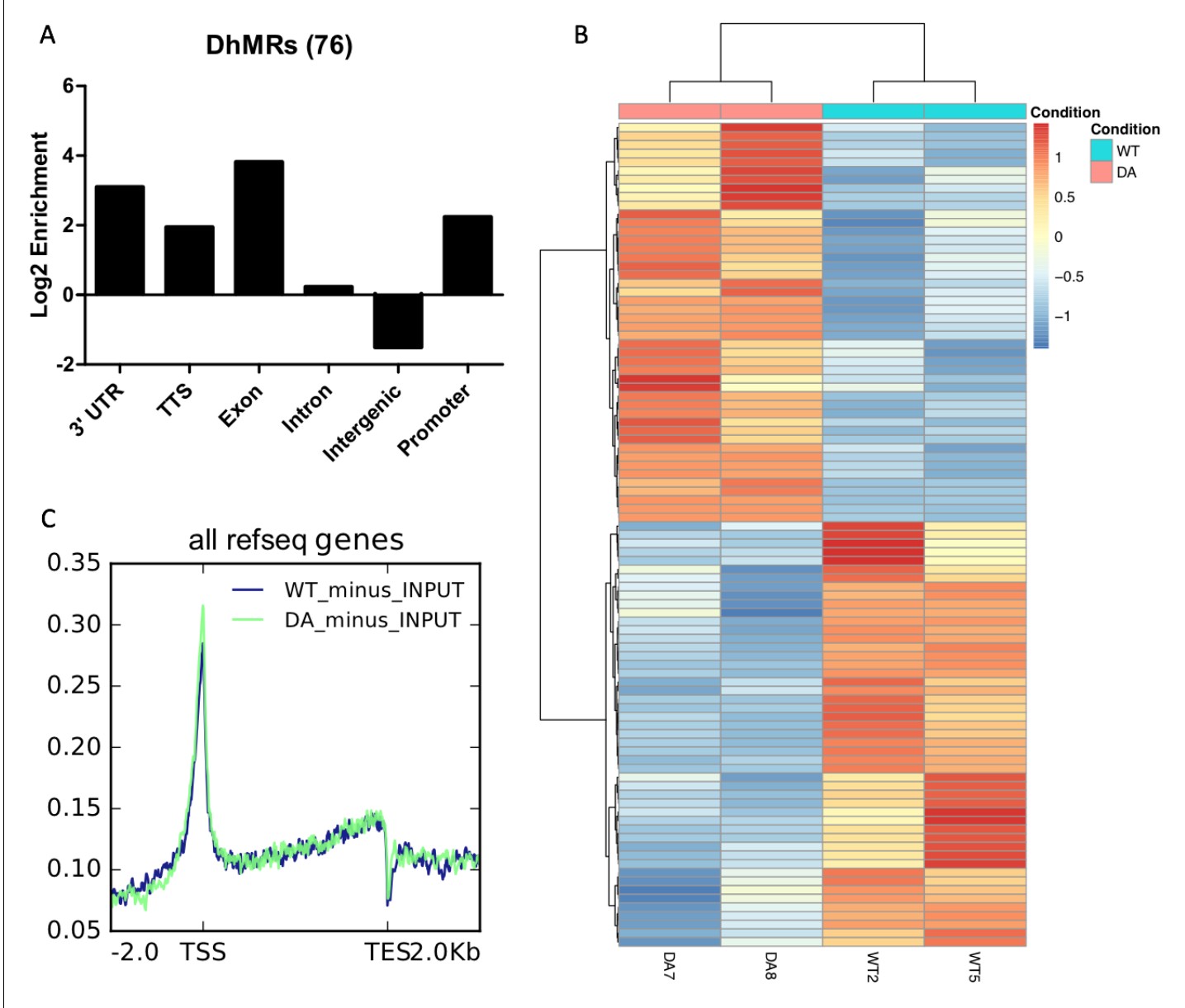

**Figure 7.** 5hmC-Seal analysis of WT and D2018A mESCs. (**A**) Genomic annotations of differentially hydroxymethylated regions in WT vs. D2018A mESCs. (**B**) Heatmap depicting 95 differentially hydroxymethylated genes in WT vs. D2018A mESCs. (**C**) Distribution of averaged hmCpG level at all genes.

DOI: https://doi.org/10.7554/eLife.34870.019

## Discussion

### A unique OGT interaction domain

We identified a 45-amino acid domain of TET1 that is both necessary and sufficient for binding of OGT. To our knowledge, this is the first time that a small protein domain has been identified that confers stable binding to OGT. The vast majority of OGT targets do not bind to OGT tightly enough to be detected in co-IP experiments, suggesting that OGT's interaction with TET proteins is unusually strong. For determination of the crystal structure of the human TET2 catalytic domain in complex with DNA, the corresponding C-terminal region was deleted (*Hu et al., 2013*), suggesting that it may be unstructured. When bound to OGT this domain may become structured, and structural

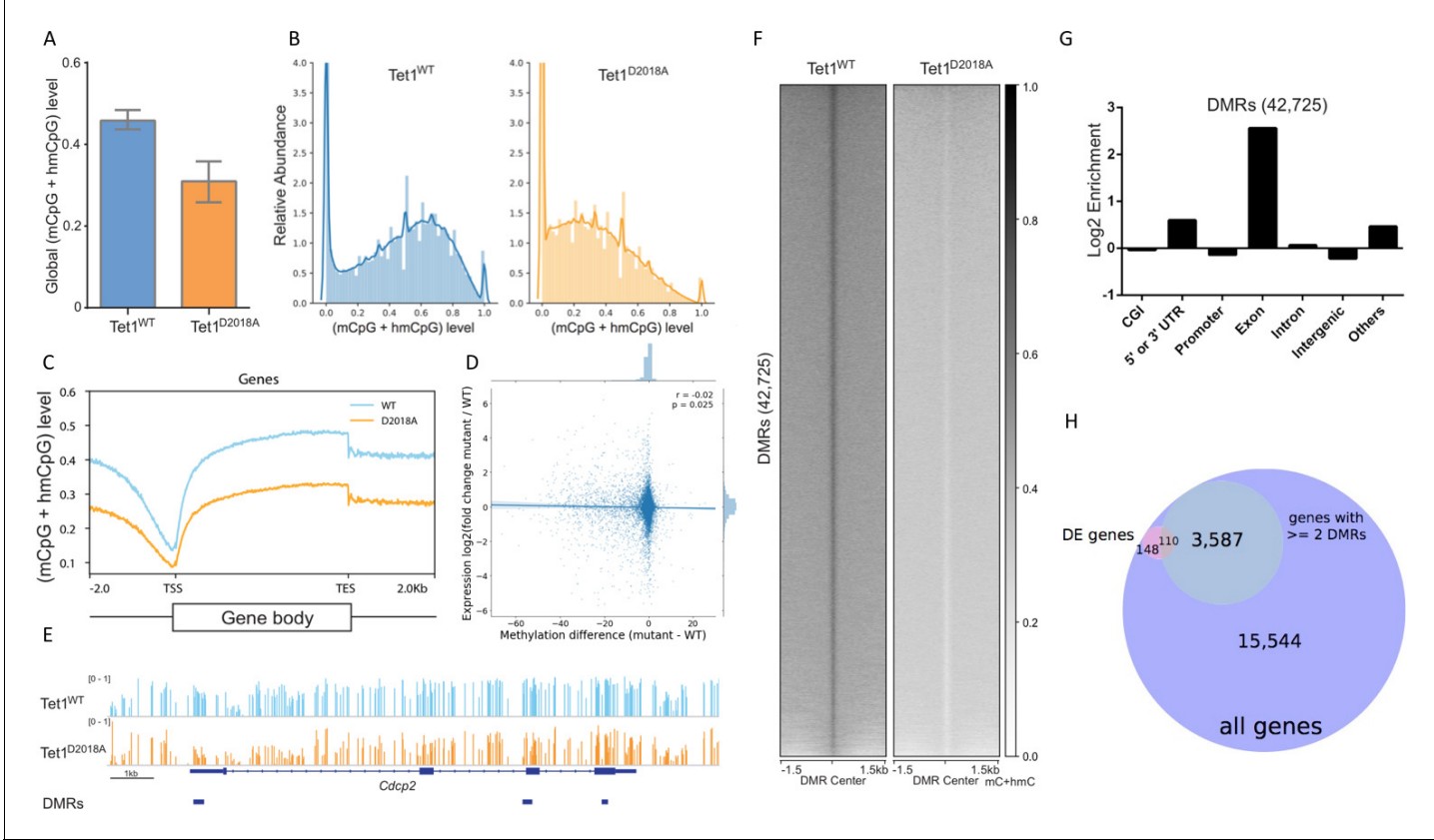

**Figure 8.** Whole genome bisulfite sequencing of WT and D2018A mESCs. (A) Genome-wide levels of mCpG +hmCpG. (B) Distribution of mC +hmC levels for individual CG sites fitting with kernel density estimate (KDE). (C) Distribution of averaged mCpG +hmCpG level at all genes. (D) Scatter plot of difference (D2018A - WT) in average CpG methylation in a 500 bp window around each promoter vs. log2 fold change (D2018A/WT) of corresponding genes. (E) An example of hypo CG-DMRs in exons of *Cdcp2* gene in D2018A mESCs. (F) Average mCpG +hmCpG level ((h)mCG/CG) of individual ranked DMRs and flanking regions (±1.5 kb). (G) Genomic annotations of hypo CG-DMRs in D2018A mESCs. CGI: CpG island. (H) Overlap between DEGs and the set of genes with ≥ 2 DMRs between the gene's start and end sites (hypergeometric p value = 1.770e-19).

DOI: https://doi.org/10.7554/eLife.34870.020

The following figure supplement is available for figure 8:

**Figure supplement 1.** Analysis of CpG modification changes in promoters of differentially expressed genes.
DOI: https://doi.org/10.7554/eLife.34870.021

studies of OGT bound to C45 could shed light on what features make this domain uniquely able to interact stably with OGT and how OGT may stimulate TET1 activity.

An alternative or additional role for the stable TET-OGT interaction may be recruitment of OGT to chromatin by TET proteins. Loss of TET1 causes loss of OGT from chromatin (*Vella et al., 2013*) and induces similar changes in transcription in both wild-type mESCs and mESCs lacking DNA methylation (*Williams et al., 2011*). This raises the possibility that TET proteins may recruit OGT to chromatin to regulate gene expression independent of 5mC oxidation. Consistent with this possibility, OGT modifies many transcription factors and chromatin regulators in mESCs (*Myers et al., 2011*) (*Figure 9*). Thus it may be that the stable TET1-OGT interaction promotes both regulation of TET1 activity by *O*-GlcNAcylation as well as recruitment of OGT to chromatin. Notably, our results show that TET1 D2018A does not rescue 5hmC levels in *tet2/3* DM zebrafish embryos to the same extent as the wild type protein, suggesting that at least part of the role of the TET1-OGT interaction in vivo is regulation of TET1 activity.

## OGT stimulation of TET activity

Our results show for the first time that OGT can modify a TET protein in vitro, and that *O*-GlcNAcyla-tion stimulates the activity of a TET protein in vitro. We have identified 8 sites of *O*-GlcNAcylation

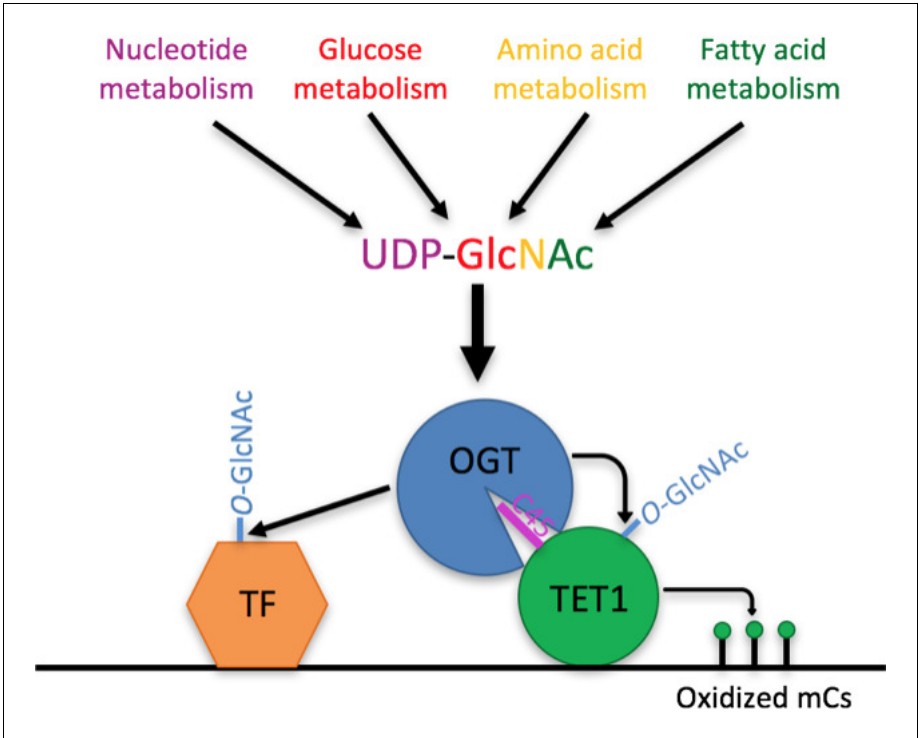

**Figure 9.** Model Model showing two roles of the TET1-OGT interaction in regulation of gene expression. OGT's activity is regulated by the abundance of its cofactor UDP-GlcNAc, whose synthesis has inputs from nucleotide, glucose, amino acid, and fatty acid metabolism. OGT (blue circle) binds to TET1 (large green circle) via the TET1 C45 (purple line). OGT modifies TET1 and regulates its catalytic activity (small green circles representing modified cytosines). At the same time, TET1 binding to DNA brings OGT into proximity of other DNA-bound transcription factors (orange hexagon), which OGT also modifies and regulates.

DOI: https://doi.org/10.7554/eLife.34870.022

within the TET1 CD (data not shown), which precludes a simple analysis of which sites are important for stimulation. Detailed studies of individual sites of modification will be required to resolve this question.

Our data are consistent with a role for OGT in TET1 regulation in cells and in vivo. OGT also directly interacts with TET2 and TET3, suggesting that it may regulate all three TET proteins. Notably, although all three TETs catalyze the same reaction, they show a number of differences that are likely to determine their biological role. Different TET proteins are expressed in different cell types and at different stages of development (*Koh et al., 2011*; *Dawlaty et al., 2011*; *Li et al., 2011*; *Zhao et al., 2015*). TET1 and TET2 appear to target different genomic regions (*Huang et al., 2014*) and to promote different pluripotent states in mESCs (*Fidalgo et al., 2016*). The mechanisms responsible for these differences are not well understood. We suggest that OGT is a strong candidate for regulation of TET enzymes.

## Regulation of TETs by OGT in development

Our result that wild type TET1 mRNA, but not TET1 mRNA carrying a mutation that can impair interaction with OGT, can rescue *tet2/3* DM zebrafish suggests that OGT regulation of TET enzymes may play a role in development. The importance of both TET proteins and OGT in development has been thoroughly established. Zebrafish lacking *tet2* and *tet3* die as larvae (*Li et al., 2015*), and knockout of *Tet* genes in mice yields developmental phenotypes of varying severities, with knockout of all three *Tet*s together being embryonic lethal (*Dawlaty et al., 2011*; *Li et al., 2011*; *Dawlaty et al., 2013*; *Dawlaty et al., 2014*). Similarly, OGT is absolutely essential for development in mice (*Shafi et al., 2000*) and zebrafish (*Webster et al., 2009*), though its vast number of targets

have made it difficult to narrow down more specifically why OGT is necessary. Our results suggest that TETs are important OGT targets in development.

## The TET1-OGT interaction regulates gene expression and DNA methylation in mESCs

The D2018A mutation reduced the TET1-OGT interaction in mESCs and altered gene expression and CpG modifications. 5hmC-Seal revealed only a small number of DhMRs and DhMGs, with 5hmC increased at about half the DhMGs and decreased at the other half. Together with mass spectrometry showing no change in total 5hmC levels, our data are consistent with a redistribution of 5hmC at a small number of genes without significantly affecting total 5hmC. The D2018A mutation is predicted to decrease TET1 activity, and the increased expression of TET2 in the D2018A cells could explain why total 5hmC levels are unchanged.

Bisulfite sequencing showed a genome-wide reduction in 5mC + 5 hmC in D2018A cells, while 5hmC-Seal revealed only a small number of DhMRs. Together these results suggest that the bisulfite sequencing largely reflects changes in 5mC. The very small correlation between promoter CpG modification and gene expression suggests that most gene expression changes cannot be explained by CpG modification differences at promoters.

The enrichment of both DMRs and DhMRs in genic regions and the correlation between gene body DMRs and DEGs suggests that changes to genic CpG modifications may underlie some of the gene expression changes. The result that OGT stimulates TET1 catalytic activity in vitro suggests that in cells the TET1 D2018A mutation should decrease TET enzyme activity, resulting in more 5mC and less 5hmC. Instead, we observed decreased 5mC and no change in 5hmC. This result is consistent with the increased expression of TET2 in D2018A mESCs compared to WT, since gene bodies are targets of TET2 rather than TET1(*Huang et al., 2014*). Higher levels of TET2 could increase 5mC oxidation at gene bodies, resulting in widespread genic demethylation and an increase in 5hmC at a small subset of genes. This result would explain the loss of 5mC in D2018A cells compared to WT and the enrichment of DMRs in genic regions.

## A connection between metabolism and the epigenome

OGT has been proposed to act as a metabolic sensor because its cofactor, UDP-GlcNAc, is synthesized via the hexosamine biosynthetic pathway (HBP), which is fed by pathways metabolizing glucose, amino acids, fatty acids, and nucleotides (*Hart et al., 2011*). UDP-GlcNAc levels change in response to flux through these pathways (*Marshall et al., 2004*; *McClain, 2002*; *Weigert et al., 2003*), leading to the hypothesis that OGT activity may vary in response to the nutrient status of the cell. Thus the enhancement of TET1 activity by OGT and the significant overlap of the two enzymes on chromatin (*Vella et al., 2013*) suggest a model in which OGT may regulate the epigenome in response to nutrient status by controlling TET1 activity (*Figure 9*).

## Materials and methods

### Cell culture

The mESC line LF2 and its derivatives were routinely passaged by standard methods in KO-DMEM, 10% FBS, 2 mM glutamine, 1X non-essential amino acids, 0.1 mM b-mercaptoethanol and recombinant leukemia inhibitory factor. HEK293T cells were cultured in DMEM, 10% FBS, and 2 mM glutamine.

The genotype of LF2 cells was confirmed by whole genome sequencing. Mycoplasma testing is performed annually. Whole genome sequencing of LF2s confirms that there is no mycoplasma contamination.

### Recombinant protein purification

Full-length human OGT in the pBJG vector was transformed into BL-21 DE3 *E. coli*. A liquid culture was grown in LB +50 µg/mL kanamycin at 37C until $OD_{600}$ reached 1.0. IPTG was added to 1 mM final and the culture was induced at 16°C overnight. Cells were pelleted by centrifugation and resuspended in 5 mL BugBuster (Novagen) +protease inhibitors (Sigma Aldrich) per gram of cell pellet. Cells were lysed on an orbital shaker for 20 min at room temperature. The lysate was clarified by

centrifugation at 30,000 g for 30 min at 4°C. Clarified lysate was bound to Ni-NTA resin (Qiagen) at 4°C and then poured over a disposable column. The column was washed with six column volumes of wash buffer 1 (20mM Tris pH 8, 1 mM CHAPS, 10% glycerol, 5 mM BME, 10 mM imidazole, 250 mM NaCl) followed by six column volumes of wash buffer 2 (wash buffer 1 with 50 mM imidazole). The protein was eluted in four column volumes of elution buffer (20 mM Tris pH 8, 1 mM CHAPS, 5 mM BME, 250 mM imidazole, 250 mM NaCl). Positive fractions were pooled and dialyzed into storage buffer (20 mM Tris pH 8, 1 mM CHAPS, 0.5 mM THP, 10% glycerol, 150 mM NaCl, 1 mM EDTA), flash frozen in liquid nitrogen and stored at −80°C in small aliquots.

Mouse TET1 catalytic domain (aa1367-2039) was expressed in sf9 insect cells according to the Bac-to-Bac Baculovirus Expression System. Constructs were cloned into the pFastBac HTA vector and transformed in DH10Bac *E. coli* for recombination into a bacmid. Bacmid containing the insert was isolated and used to transfect adherent sf9 cells for 6 days at 25°C. Cell media (P1 virus) was isolated and used to infect 20 mL of sf9 cells in suspension for 3 days. Cell media (P2 virus) was isolated and used to infect a larger sf9 suspension culture for 3 days. Cells were pelleted by centrifugation, resuspended in lysis buffer (20 mM Tris pH 8, 1% Triton, 10% glycerol, 20 mM imidazole, 50 mM NaCl, 1 mM MgCl$_2$, 0.5 mM TCEP, protease inhibitors, 2.5 U/mL benzonase), and lysed by douncing and agitation at 4C for 1 hr. The lysate was clarified by centrifugation at 48,000 g for 30 min at 4°C and bound to Ni-NTA resin (Qiagen) at 4°C, then poured over a disposable column. The column was washed with five column volumes of wash buffer (20 mM Tris pH 8, 0.3% Triton, 10% glycerol, 20 mM imidazole, 250 mM NaCl, 0.5 mM TCEP, protease inhibitors). The protein was eluted in five column volumes of elution buffer (20 mM Tris pH 8, 250 mM imidazole, 250 mM NaCl, 0.5 mM TCEP, protease inhibitors). Positive fractions were pooled and dialyzed overnight into storage buffer (20 mM Tris pH 8, 150 mM NaCl, 0.5 mM TCEP). Dialyzed protein was purified by size exclusion chromatography on a 120 mL Superdex 200 column (GE Healthcare) in storage buffer. Positive fractions were pooled, concentrated, flash frozen in liquid nitrogen and stored at −80C in small aliquots.

## Overexpression in HEK293T cells and immunoprecipitation

Mouse Tet1 catalytic domain (aa1367-2039) and truncations and mutations thereof were cloned into the pcDNA3b vector. GFP fusion constructs were cloned into the pcDNA3.1 vector. Human OGT constructs were cloned into the pcDNA4 vector. Plasmids were transiently transfected into adherent HEK293T cells at 70 − 90% confluency using the Lipofectamine 2000 transfection reagent (Thermo-Fisher) for 1–3 days.

Transiently transfected HEK293T cells were harvested, pelleted, and lysed in IP lysis buffer (50 mM Tris pH 8, 200 mM NaCl, 1% NP40, 1x HALT protease/phosphatase inhibitors). For pulldown of FLAG-tagged constructs, cell lysate was bound to anti-FLAG M2 magnetic beads (Sigma Aldrich) at 4°C. For pulldown of GFP constructs, cell lysate was bound to magnetic protein G dynabeads (ThermoFisher) conjugated to the JL8 GFP monoclonal antibody (Clontech) at 4°C. Beads were washed three times with IP wash buffer (50 mM Tris pH 8, 200 mM NaCl, 0.2% NP40, 1x HALT protease/phosphatase inhibitors). Bound proteins were eluted by boiling in SDS sample buffer.

## In vitro transcription/translation and immunoprecipitation

GFP fused to TET C-terminus peptides were cloned into the pcDNA3.1 vector and transcribed and translated in vitro using the TNT Quick Coupled Transcription/Translation System (Promega).

For immunoprecipitation, recombinant His-tagged OGT was coupled to His-Tag isolation dynabeads (ThermoFisher). Beads were bound to in vitro translation extract diluted 1:1 in binding buffer (40 mM Tris pH 8, 200 mM NaCl, 40 mM imidazole, 0.1% NP40) at 4°C. Beads were washed 3 times with wash buffer (20 mM Tris pH 8, 150 mM NaCl, 20 mM imidazole, 0.1% NP40). Bound proteins were eluted by boiling in SDS sample buffer.

## Recombinant protein binding assay

20 µL reactions containing 2.5 µM rOGT and 2.5 µM rTET1 CD wt or D2018A were assembled in binding buffer (50 mM Tris pH 7.5, 100 mM NaCl, 0.02% Tween-20) and pre-incubated at room temperature for 15 min. TET1 antibody (Millipore 09 − 872) was bound to magnetic Protein G Dynabeads (Invitrogen), and beads were added to reactions following pre-incubation. Reactions were bound to beads for 10 min at room temperature. Beads were washed 3 times with 100 µL binding

buffer, and bound proteins were recovered by boiling in SDS sample buffer and analyzed by SDS-PAGE and coomassie stain.

## Western blots

For western blot, proteins were separated on a denaturing SDS-PAGE gel and transferred to PVDF membrane. Membranes were blocked in PBST +5% nonfat dry milk at room temp for >10 min or at 4°C overnight. Primary antibodies used for western blot were: FLAG M2 monoclonal antibody (Sigma Aldrich F1804), TET2 monoclonal antibody (Millipore MABE462), TET3 polyclonal antibody (Millipore ABE383), OGT polyclonal antibody (Santa Cruz sc32921), OGT monoclonal antibody (Cell Signaling D1D8Q), His6 monoclonal antibody (Thermo MA1-21315), JL8 GFP monoclonal antibody (Clontech), and O-GlcNAc RL2 monoclonal antibody (Abcam ab2739). Secondary antibodies used were goat anti-mouse HRP and goat anti-rabbit HRP from BioRad. Blots were incubated with Pico Chemiluminescent Substrate (ThermoFisher) and exposed to film in a dark room.

## Slot blot

DNA samples were denatured in 400 mM NaOH +10 mM EDTA by heating to 95°C for 10 min. Samples were placed on ice and neutralized by addition of 1 vol of cold NH$_4$OAc pH 7.2. DNA was loaded onto a Hybond N + nylon membrane (GE) by vacuum using a slot blot apparatus. The membrane was dried at 37°C and DNA was covalently linked to the membrane by UV crosslinking (700uJ/cm$^2$ for 3 min). Antibody binding and signal detection were performed as outlined for western blotting using 5hmC monoclonal antibody (Active Motif 39791).

For the loading control, membranes were analyzed using the Biotin Chromogenic Detection Kit (Thermo Scientific) according to the protocol. Briefly, membranes were blocked, probed with streptavidin conjugated to alkaline phosphatase (AP), and incubated in the AP substrate BCIP-T (5-bromo-4-chloro-3-indolyl phosphate, p-toluidine salt). Cleavage of BCIP-T causes formation of a blue precipitate.

For quantification of slot blots, at least three biological replicates were used. Signal was normalized to the loading control and significance was determined using the unpaired t test.

## Preparation of lambda DNA substrate

Linear genomic DNA from phage lambda (dam-, dcm-) containing 12 bp 5′ overhangs was purchased from Thermo Scientific. Biotinylation was performed by annealing and ligating a complementary biotinylated DNA oligo. Reactions containing 175 ng/µL lambda DNA, 2 µM biotinylated oligo, and 10 mM ATP were assembled in 1x T4 DNA ligase buffer, heated to 65°C, and cooled slowly to room temperature to anneal. 10uL T4 DNA ligase was added and ligation was performed overnight at room temperature. Biotinylated lambda DNA was purified by PEG precipitation. To a 500 µL ligation reaction, 250 µL of 60% PEG8000 +10 mM MgCl$_2$ was added and reaction was incubated at 4°C overnight with rotation. The next day DNA was pelleted by centrifugation at 14,000 g at 4°C for 5 min. Pellet was washed with 1 mL of 75% ethanol and resuspended in TE.

Biotinylated lambda DNA was methylated using M.SssI CpG methyltransferase from NEB. 20 µL reactions containing 500 ng lambda DNA, 640 µM S-adenosylmethionine, and four units methyltransferase were assembled in 1x NEBuffer 2 supplemented with 20 mM Tris pH eight and incubated at 37°C for 1 hr. Complete methylation was confirmed by digestion with the methylation-sensitive restriction enzyme BstUI from NEB.

## In vitro TET1 CD O-GlcNAcylation

In vitro modification of rTET1 CD with rOGT was performed as follows: 10 µL reactions containing 1 µM rTET1 CD, 1 – 5 µM rOGT, and 1 mM UDP-GlcNAc were assembled in reaction buffer (50 mM HEPES pH 6.8, 150 mM NaCl, 10% glycerol, 0.5 mM TCEP) and incubated at 37C for 30 – 60 min or at 4°C for 18 – 24 hr.

## In vitro TET1 CD activity assays

20 µL reactions containing 100 ng biotinylated, methylated lambda DNA, rTET1 CD (from frozen aliquots or from in vitro O-GlcNAcylation reactions), and TET cofactors (1 mM alpha-ketoglutarate, 2 mM ascorbic acid, 100 uM ferrous ammonium sulfate) were assembled in reaction buffer (50 mM

HEPES pH 6.8, 100 mM NaCl) and incubated at 37°C for 10 – 60 min. Reactions were stopped by addition of 1 vol of 2M NaOH +50 mM EDTA and DNA was analyzed by slot blot.

Generation of mouse embryonic stem cell lines mESC lines (*Figure 6—figure supplement 1*) were derived using CRISPR-Cas9 genome editing. A guide RNA to the Tet1 3'UTR was cloned into the px459-Cas9-2A-Puro plasmid using published protocols (*Ran et al., 2013*) with minor modifications. Templates for homology directed repair were amplified from Gene Blocks (IDT) (*Supplementary file 2* and *3*). Plasmid and template were co-transfected into LF2 mESCs using FuGENE HD (Promega) according to manufacturer protocol. After two days cells were selected with puromycin for 48 hr, then allowed to grow in antibiotic-free media. Cells were monitored for green or red fluorescence (indicating homology directed repair) and fluorescent cells were isolated by FACS 1 – 2 weeks after transfection. All cell lines were propagated from single cells and correct insertion was confirmed by PCR genotyping (*Figure 6—figure supplement 1*, *Supplementary file 2*).

Genome-wide profiling of 5mC and 5hmC revealed a 25 kb deletion in WT but not D2018A cells distal to the CRISPR/Cas9 cut site (*Figure 6—figure supplement 3A,B*). A recent report shows that large deletions like this are more common than previously appreciated(*Kosicki et al., 2018*). Analysis of cells with wild-type TET1 and one intact copy of this region shows that the differences in gene expression and 5mC levels are caused by the TET1 D2018A mutation rather than the 25 kb deletion (*Figure 6—figure supplement 3C,D*, *Figure 6—figure supplement 3—source datas 1 and 2*).

## Nucleotide mass spectrometry

Genomic DNA (3 μg) was subjected to hydrolysis with PDE I (3.6U), PDE II (3.2U), DNase I (50U), and alkaline phosphatase (10U) in 10 mM Tris HCl/15 mM MgCl$_2$ buffer (pH 7) at 37°C overnight. The hydrolysates were spiked with $^{13}C_{10}^{15}N_2$-5-methyl-2'-deoxycytidine (1 pmol) and 5-hydroxymethyl-d$_2$-2'-deoxycytidine-6-d$_1$ (500 fmol) (internal standards for mass spectrometry) and filtered through Nanosep 10K Omega filters (Pall Corporation, Port Washington, NY).

Quantitation of mC and hmC was performed using a Dionex Ultimate 3000UHPLC (Thermo Fisher, Waltham MA) interfaced with a Thermo TSQ Vantage mass spectrometer (Thermo Fisher). Chromatographic separation was achieved on a Luna Omega Polar C18 column (150 × 1.0 mm, 1.6 μm, Phenomenex, Torrance CA) heated to 50°C and eluted at a flow rate of 50 μL/min with a gradient of 0.1% acetic acid in H$_2$O (A) and acetonitrile (B). A linear gradient of 1% to 5% B in 5.7 min was used, followed by an increase to 20% B over 1.1 min and a further increase to 50% B in 1.1 min. Solvent composition was a returned to initial conditions (1% B) and the column was re-equilibrated for 7 min. Under these conditions, mC and $^{13}C_{10}^{15}N_2$-MeC eluted at 3.7 min, both hmC and the internal standard D$_3$-hmC eluted at 2.9 min. Quantitation was achieved by monitoring the transitions $m/z$ 258.2 [M + H$^+$] → $m/z$ 142.1 [M – deoxyribose + H$^+$] for hmC, $m/z$ 261.2 [M + H$^+$] → $m/z$ 145.1 [M – deoxyribose + H$^+$] for D$_3$-hmC, $m/z$ 242.1 [M + H$^+$] → $m/z$ 126.1 [M + H$^+$] for mC, $m/z$ 254.2 [M + H$^+$] → $m/z$ 133.1 [M + H$^+$] for $^{13}C_{10}^{15}N_2$-mC. Optimal mass spectrometry conditions were determined by infusion of authentic standards. Typical settings on the mass spectrometer were: a spray voltage of 3500 V, a sheath gas of 12 units, the declustering voltage was 5 V, the RF lens was 55 V, the vaporizer temperature was 75 °C, and the ion transfer tube was maintained at 350 °C. The full-width at half-maximum (FWHM) was maintained at 0.7 for both Q1 and Q3. Fragmentation was induced using a collision gas of 1.5 mTorr and a collision energy of 10.3 V for mC and 10.6 V for hmC.

## RNA-seq

RNA was extracted from mESCs (two biological replicates each for WT and D2018A, and two technical replicates for each biological replicate) using Zymo DirectZol RNA miniprep kit. 200 ng of RNA per sample was used for Lexogen Ribocop rRNA depletion. Libraries were prepared from 8 μL of Ribocop-treated RNA using Lexogen SENSE Total RNA-seq Library Prep Kit.

Libraries were sequenced on an Illumina Hiseq 4000 with single-end 50 base reads. Reads were aligned to the mouse genome (Ensembl build GRCm38.p6) and gene counts were created using STAR_2.5.3a. Normalization and differential expression analysis was performed with DESeq2 v1.20.0. Data was visualized using Matplotlib.

## Methyl-seq library preparation and whole genome bisulfite sequencing

Genomic DNA was extracted from 1 million mESCs (two biological replicates each for wild type and Tet1 D2018A) using the DNeasy kit from Qiagen. Methyl-Seq libraries were prepared using Accel-NGS Methyl-Seq DNA Library Kit with 30 ng gDNA for each sample and were sequenced on Illumina HiSeq 4000 with 150 bases pair-ended reads. For data processing, the raw reads were first trimmed to remove Illumina adapters and PCR duplicates and then mapped to mm10 mouse reference genome using Bismark (https://www.bioinformatics.babraham.ac.uk/projects/bismark/). The alignment files (BAM file) generated were analyzed by MethylPy (https://github.com/yupenghe/methylpy) (*Lister et al., 2013*; *He, 2018*) to get the methylation level at individual cytosines. Generally, the methylation level is defined as the ratio of the sum of methylated basecall counts over the total basecall counts at each individual pairwise cytosine on both strands. The significantly methylated cytosine sites (DMSs) were identified using a binomial test for each CpG context with FDR < 0.01 as described previously(*Ma et al., 2014*). Differentially methylated regions (DMRs) were defined by the DMRfind function in MethylPy by joining at least two DMSs within 250 bp.

## 5hmC-Seal

5hmC profiling was performed as described(*Han et al., 2016*). Briefly, 100 ng genomic DNA were fragmented in Tagmentation buffer at 55°C. Fragmented DNA was purified by Zymo DNA Clean and Concentration Kit. Then, the selective 5hmC chemical labeling was performed in glucosylation buffer (50 mM HEPES buffer pH 8.0, 25 mM MgCl$_2$) containing above fragmented DNA, βGT, N$_3$-UDP-Glc, and incubated at 37°C for 2 hr. After DNA purification in ddH$_2$O, DBCO-PEG4-Biotin (Click Chemistry Tools) was added and incubated at 37°C for 2 hr. The biotin labeled DNA was pulled down by C1 Streptavidin beads (Life Technologies) for 15 min at room temperature. Next, the captured DNA fragments were subjected to PCR amplification using Nextera DNA sample preparation kit. The resulting amplified product was purified by 1.0X AMPure XP beads. Input library was made by direct PCR from fragmented DNA directly without labeling and pull-down. The libraries were quantified by a Qubit fluorometer (Life Technologies) and sequenced on NextSeq 500 PE42.

Adaptors and low quality nucleotides were trimmed from raw sequencing reads by Trim_Galore, and bowtie was used to align clean reads to mm9 reference genome. Peak calling was performed by MACS1.4. DESeq2 was further used to calculate differentially hydroxymethylated genes and regions.

## RT-qPCR

Total RNA was isolated from mESCs using Direct-zol RNA miniprep kit from Zymo. 1 µg of RNA was used for cDNA synthesis using the iScript Reverse Transcription kit from BioRad. cDNA was used for qPCR using the SensiFAST SYBR Lo-Rox kit from Bioline. Relative gene expression levels were calculated using the $\Delta\Delta C_t$ method. See *Supplementary file 4* for primer sequences.

## Zebrafish mRNA rescue experiments

Zebrafish husbandry was conducted under full animal use and care guidelines with approval by the Memorial Sloan-Kettering animal care and use committee. For mRNA rescue experiments, mTET1D2018A and mTET1wt plasmids were linearized by NotI digestion. Capped RNA was synthesized using mMessage mMachine (Ambion) with T7 RNA polymerase. RNA was injected into one-cell-stage embryos derived from tet2$^{mk17/mk17}$, tet3$^{mk18/+}$ intercrosses at the concentration of 100 pg/embryo (*Li et al., 2015*). Injected embryos were raised under standard conditions at 28.5°C until 30 hr post-fertilization (hpf) at which point they were fixed for in situ hybridization using an antisense probe for *runx1*. The *runx1* probe is described in (*Kalev-Zylinska et al., 2002*); in situ hybridization was performed using standard methods, and runx1 levels were scored across samples without knowledge of the associated experimental conditions (*Thisse and Thisse, 2008*). Examples of larvae categorized as runx1 high and runx1 low are provided in *Supplementary file 5*. tet2/3 double mutants were identified based on morphological criteria and mutants were confirmed by PCR genotyping after in situ hybridization using previously described primers (*Li et al., 2015*).

For sample size estimation for rescue experiments, we assume a background mean of 20% positive animals in control groups. We anticipate a significant change would result in at least a 30% difference between the experimental and control means with a standard deviation of no more than 10. Using the 1-Sample Z-test method, for a specified power of 95% the minimum sample size is 4.

Typically, zebrafish crosses generate far more embryos than required. Experiments are conducted using all available embryos. The experiment is discarded if numbers for any sample are below this minimum threshold when embryos are genotyped at the end of the experimental period. Injections were separately performed on clutches from five independent crosses; p values are based on these replicates and were derived from the unpaired two-tailed t test. Embryo numbers for all five biological replicates are included in *Supplementary file 5*.

For the dot blot, genomic DNA was isolated from larvae at 30hpf by phenol-chloroform extraction and ethanol precipitation. Following RNase treatment and denaturation, 2-fold serially diluted DNA was spotted onto nitrocellulose membranes. Cross-linked membranes were incubated with 0.02% methylene blue to validate uniform DNA loading. Membranes were blocked with 5% BSA and incubated with anti-5hmC antibody (1:10,000; Active Motif) followed by a horseradish peroxidase-conjugated antibody (1:15,000; Active Motif). Signal was detected using the ECL Prime Detection Kit (GE). The results of three independent experiments were quantified using ImageJ at the lowest dilution and exposure where signal was observed in Tet1 injected embryos. To normalize across blots, all values are presented as the ratio of 5hmC signal in experimental animals divided by wild-type control signal from the same blot.

## Reproducibility and Rigor

All IP-Westerns are representative of at least three independent biological replicates (experiments carried out on different days with a different batch of HEK293T cells or mESCs). For targeted mESC lines, three independently derived lines for each genotype were assayed in at least two biological replicates. For in vitro activity and binding assays using recombinant proteins (representing multiple protein preparations), data represent at least three technical replicates (carried out on multiple days). The zebrafish rescue experiment was performed five times (biological replicates), with dot blots carried out three times. We define an outlier as a result in which all the controls gave the expected outcome but the experimental sample yielded an outcome different from other biological or technical replicates. There were no outliers or exclusions.

## Acknowledgments

We thank Miguel Ramalho-Santos for the FLAG-TET1 CD plasmid and Suzanne Walker for the His-OGT plasmid. We thank Richard Yan, Myles Hochman, and Sy Redding for technical assistance. We thank all members of the Panning lab for valuable ideas and discussion. This work was supported by R01 GM088506 (BP), NCI grant P30 CA008748 (MG), and funding from the Geoffrey Beene Cancer Research Center of Memorial Sloan-Kettering Cancer Center (MG). JH was supported by the California Institute for Regenerative Medicine Predoctoral Fellowship TG2-01153.

## Additional information

### Funding

| Funder | Grant reference number | Author |
| --- | --- | --- |
| National Cancer Institute | P30 CA008748 | Cheng Li<br>Mary Goll |
| California Institute for Regenerative Medicine | TG2-01153 | Joel Hrit<br>Barbara Panning |
| National Institutes of Health | R01 GM088506 | Cheng Li<br>Mary Goll |
| Geoffrey Beene Cancer Research Center of Memorial Sloan-Kettering Cancer Center | | Mary Goll |

The funders had no role in study design, data collection and interpretation, or the decision to submit the work for publication.

## Author contributions

Joel Hrit, Mary Goll, Barbara Panning, Conceptualization, Data curation, Formal analysis, Supervision, Funding acquisition, Investigation, Methodology, Writing—original draft, Project administration, Writing—review and editing; Leeanne Goodrich, Cheng Li, Conceptualization, Data curation, Formal analysis, Investigation, Methodology, Writing—original draft, Writing—review and editing; Bang-An Wang, Ji Nie, Xiaolong Cui, Data curation, Formal analysis, Investigation, Methodology, Writing—original draft, Writing—review and editing; Elizabeth Allene Martin, Eric Simental, Monica Yun Liu, Data curation, Formal analysis, Investigation, Methodology; Jenna Fernandez, Data curation, Formal analysis, Writing—original draft, Writing—review and editing; Joseph R Nery, Rosa Castanon, Data curation, Formal analysis; Rahul M Kohli, Natalia Tretyakova, Joseph R Ecker, Formal analysis, Methodology, Project administration, Writing—review and editing; Chuan He, Data curation, Formal analysis, Project administration, Writing—review and editing

## Author ORCIDs

Joel Hrit (iD) http://orcid.org/0000-0002-4497-956X
Leeanne Goodrich (iD) http://orcid.org/0000-0003-0603-4503
Bang-An Wang (iD) http://orcid.org/0000-0003-4488-1738
Eric Simental (iD) http://orcid.org/0000-0002-8638-6578
Monica Yun Liu (iD) https://orcid.org/0000-0003-3936-9377
Natalia Tretyakova (iD) http://orcid.org/0000-0002-0621-6860
Chuan He (iD) http://orcid.org/0000-0003-4319-7424
Joseph R Ecker (iD) http://orcid.org/0000-0001-5799-5895
Mary Goll (iD) http://orcid.org/0000-0001-5003-6958
Barbara Panning (iD) https://orcid.org/0000-0002-8301-1172

## Decision letter and Author response

Decision letter https://doi.org/10.7554/eLife.34870.034
Author response https://doi.org/10.7554/eLife.34870.035

# Additional files

## Supplementary files

• Supplementary file 1. Genes changed 2-fold or more in D2018A vs WT mESCs by RNA-seq
DOI: https://doi.org/10.7554/eLife.34870.023

• Supplementary file 2. Primers used for creating and genotyping mESC lines
DOI: https://doi.org/10.7554/eLife.34870.024

• Supplementary file 3. Gene blocks amplified to make HDR templates
DOI: https://doi.org/10.7554/eLife.34870.025

• Supplementary file 4. Primers used for qPCR
DOI: https://doi.org/10.7554/eLife.34870.026

• Supplementary file 5. Analysis of zebrafish larvae. (A) Representative images of larvae with high and low *runx1* expression. (B) Embryo numbers and scoring for all five biological replicates.
DOI: https://doi.org/10.7554/eLife.34870.027

• Transparent reporting form
DOI: https://doi.org/10.7554/eLife.34870.028

## Data availability

5hmC-Seal data has been uploaded to GEO under accession GSE119500. High throughput RNA-seq and WGBS data has been uploaded to GEO under accession GSE119666.

The following datasets were generated:

| Author(s) | Year | Dataset title | Dataset URL | Database and Identifier |
| --- | --- | --- | --- | --- |
| Nie J, Cui X, Hrit J, Panning B, He C | 2018 | OGT binds a conserved C-terminal domain of TET1 to regulate TET1 | https://www.ncbi.nlm.nih.gov/geo/query/acc. | NCBI Gene Expression Omnibus, |

| | | | activity and function in development | cgi?acc=GSE119500 | GSE119500 |
|---|---|---|---|---|---|
| Wang B, Hrit J, Nery J, Castanon R, Panning B, Ecker JR | | 2018 | Perturbation of the OGT-TET1 interaction in mouse embryonic stem cells | https://www.ncbi.nlm.nih.gov/geo/query/acc.cgi?acc=GSE119666 | NCBI Gene Expression Omnibus, GSE119666 |

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
