## [Decision Letter]

Thank you for submitting your article "OGT binds a conserved C-terminal domain of TET1 to regulate TET1 activity and function in development" for consideration by *eLife*. Your article has been reviewed by three peer reviewers, and the evaluation has been overseen by a Reviewing Editor and Kevin Struhl as the Senior Editor. The reviewers have opted to remain anonymous. *eLife*

The reviewers have discussed the reviews with one another and the Reviewing Editor has drafted this decision to help you prepare a revised submission.

Summary:

This manuscript identifies a region of the TET1 protein that interacts with the OGT enzyme and demonstrates that OGT-catalyzed modification stimulates TET1 activity in vitro. The authors show that a TET1 mutation (D2018A) that disrupts interaction with OGT prevents TET1 from rescuing hematopoietic stem cell production and 5-hydroxymethylcytosine (5hmC) levels in *tet* mutant zebrafish embryos, suggesting that OGT stimulates TET1 activity in vivo. The authors also found that the D2018A mutation alters the abundance and composition of TET-containing molecular complexes in mouse embryonic stem cells, and that this mutation alters gene expression and the distribution of 5-methylcytosine (5mC).

Essential revisions:

The reviewers and editors feel that the biological function of the interaction between OGT and TET1 in ESCs requires further investigation, as the presented phenotypes are complex and somewhat confusing. Specifically:

1) ES cells with homozygous TET1 D2018A mutation have significantly elevated levels of TET2, as well as a higher proportion of TET2 that co-fractionates with OGT and HCF1 within high molecular weight complexes (Figure 6). The elevated levels of TET2 in TET1 D2018A mutant cells may reflect increased stability through formation of large complexes with OGT, HCF1, and potentially other proteins. Although TET2 function is not the primary focus of the study, the phenotype of TET1 D2018A mutant cells, in which 5mC is redistributed and seemingly reduced while 5hmC levels are unchanged, merits further examination. The authors should test whether the stability or expression of TET2 is increased in D2018A mutant cells.

2) The images in Figure 7D suggest that ES cells with the TET1 D2018A mutation have reduced 5mC levels, but the immunofluorescence staining only gives limited information on how TET1-OGT interaction affects the distribution of 5mC and 5hmC. The authors conclude that the distribution of 5mC is altered, but the overall levels are not directly measured, and the interpretation that 5hmC is similar between mutant and WT, but that 5mC (the substrate) is redistributed and potentially reduced is confusing. Changes in bulk 5mC levels should be measured in these cells. The authors should also conduct genome-wide profiling of 5mC and 5hmC, and potentially of TET1, TET2, TET3 and OGT chromatin binding.

3) Although reduction of TET1 activity would not be predicted to cause reduction of 5mC levels on DNA, the authors suggest increased TET2 activity may compensate for reduced TET1 activity in D2018A ES cells. Direct measurement of the 5mC oxidation activities of TET1 and TET2 complexes in wild type and D2018A cells would be difficult, but the authors may be able to restore normal levels and distribution of 5mC in D2018A cells by overexpression of the GFP-C45 fusion, which binds OGT in vitro (Figure 3). This would provide evidence that the OGT interaction domain of TET1 has two important functions, activation of TET1 activity and titration of OGT to limit activation of TET2. Knockdowns of TET2 and/or TET3 in the WT and mutant cells could also isolate the effects of the TET1 mutation.

4) The authors should add more analysis to the RNA-seq data. For example, how does the set of differentially expressed genes overlap with OGT ChIP-seq targets in published studies?

[Editors’ note: a previous version of this study was rejected after peer review, but the authors submitted for reconsideration. The first decision letter after peer review is shown below.]

Thank you for submitting your work entitled "OGT binds a conserved C-terminal domain of TET1 to regulate TET1 activity and function in development" for consideration by *eLife*. Your article has been reviewed by three peer reviewers, and the evaluation has been overseen by a Reviewing Editor and a Senior Editor. The reviewers have opted to remain anonymous.

Our decision has been reached after consultation between the reviewers. Based on these discussions and the individual reviews below, we regret to inform you that your work in its current form will not be considered further for publication in *eLife*.

The reviewers and editors found regulation of TET1 activity by OGT-catalyzed *O*-GlcNAc modification to be potentially very interesting, but a number of serious concerns were raised about the validity of the conclusions in this study. Perhaps the most significant finding is that *O*-GlcNAc modification of TET1 by OGT strongly stimulates its activity, however the enzymatic activity of the D2018 mutant is similar to that of wild type TET1 in 293T cells, which contain endogenous OGT. The in vivo results therefore are apparently inconsistent with the in vitro results. The data in the paper also do not clearly establish whether *O*-GlcNAc modification, rather than OGT binding, affects TET1 activity. The finding that all the TPR motifs of OGT are important for the interaction with TET1 raised concerns because different TPR motifs usually interact with different OGT substrates. The interpretations – and interpretability – of several key experiments were also questioned by the reviewers.

Overall, given the published link between OGT and TET enzymes, we would expect a manuscript to present a major advance on the biological significance and/or mechanism of this interaction. If the activation of TET1 by OGT-catalyzed *O*-GlcNAc modification can be unambiguously established in vitro and in vivo, and the biological function and/or mechanism substantively explored, we would be happy to consider a revised manuscript as a new submission.

*Reviewer #1:*

TET enzymes oxidize 5mC towards cytosine demethylation. In this manuscript, the authors characterized the interaction between TET1 and OGT. They showed that C-terminus of TET1 directly interacted with OGT. This interaction might be required for activity of TET1 in vivo. Overall, the authors proposed and tested a novel hypothesis on the interaction between OGT and TET1. However, it is very confusing to see that the D2018A mutant of TET1 lost the interaction with OGT but still retained enzymatic activity in 293T cells. A careful and through study is needed to justify the major conclusions in this manuscript.

1) It is unclear if OGT is required for the enzymatic activity of TET1 in mammalian cells. The authors showed that the D2018A mutant was still able to convert 5mC to 5hmC in 293T cells. The enzymatic activity of the D2018 mutant was similar to that of wild type TET1 (Figure 5B and C). As the expression of OGT is very high in 293T cells, it indicates that without the interaction with OGT, TET1 is still able to oxidize 5mC, which contradicts with the major conclusions of the manuscript.

2) The authors stated "we mutated these residues individually and found that D2018A eliminated detectable interaction between FLAG-tagged TET1 CD and myc-OGT". However, in the Figure 3D, only D2018A had been examined. It is unclear if other mutations abolish the interaction between TET1 and OGT. In particular, D2018 is not mutated in TET1 CD mt2. But TET1 CDmt2 still lost the interaction with OGT, indicating that other residues such as V2021 and T2022 may play more important roles for the interaction.

3) The authors have not demonstrated if the enzymatic activity of OGT is required for TET1-dependent functions. The authors need to show if the enzymatic dead OGT is able to activate TET1 both in vitro and in vivo.

4) The study on the TET1 mutant is incomplete. It is unclear if OGT is able to promote the enzymatic activity of the mutant TET1 in vitro.

5) The authors showed a conserved interacting motif existed in both TET2 and TET3. Does TET2 or TET3 interact with OGT via the same motif? Does OGT promote TET2 and TET3 as well?

6) It is very strange to see that all the TPR motifs of OGT are important for the interaction. Usually, different TPR motifs interact with different OGT substrates. If all the TPR repeats are required for the interaction with TET1, will other OGT substrates, such as HIF-1, affect the interaction between OGT and TET1? Especially HIF-1 is in the same complex with TET1 and OGT. Is the catalytic domain of OGT also required for the interaction with TET1?

Reviewer #2:

Hrit et al. follow up on work from several groups showing *O*-GlcNAc transferase (OGT) binds to TET proteins, and find that *O*-GlcNAc modification of TET1 stimulates its methylcytosine oxidation activity in vitro. In addition, the authors perform extensive mapping of the domain of TET1 that binds OGT, and find a 45 amino acid region of the TET1 catalytic domain that is necessary and sufficient for this interaction. Finally, the authors demonstrate that mRNA corresponding to wild type TET1, but not D2018A mutant TET1, can partially rescue HSC development in zebrafish lacking *tet2* and *tet3*, as well as overall hmC levels. These findings contrast with reports that *O*-GlcNAc modification of TET1 increases Tet1 stability (PMID: 23729667), but have no effect on its catalytic activity (PMID: 24394411).

Several studies have established that TET proteins and OGT broadly regulate each other's functions. However, there are conflicting data in the literature regarding the mechanisms by which this occurs. Consequently, the question is important and these findings contribute a new potential mechanism by which OGT regulates TET1. However, the extent to which the D2018A mutant prevents *O*-GlcNAc modification and stimulation of TET1 activity by OGT is not thoroughly tested. In addition, the study would be strengthened by a deeper analysis of how TET1's activity is regulated by *O*-GlcNAc modification (see below).

1) Although there is a non-quantitative hmC blot that shows reduced hmC in fish rescued with mutant TET1 mRNA (Figure 6C), it is not clear whether reduction of meC oxidation is due to reduced *O*-GlcNAc modification, reduced TET1/OGT physical interaction, or both. The relative contributions of TET1/OGT interaction and *O*-GlcNAc modification on TET1 activity could be measured in the authors' in vitro system.

2) Perhaps the most interesting new finding is the demonstration that *O*-GlcNAc modification of TET1 by OGT strongly stimulated its activity (Figure 1B and C). This observation would be strengthened by examination of how *O*-GlcNAc modification leads to enhanced catalytic activity. Does this modification increase the affinity of TET1 for meC? Does *O*-GlcNAc affect the Km or Vmax of the enzyme? These questions should be testable with existing reagents used in Figure 1.

*Reviewer #3:*

Hrit et al. map and characterize the binding site of OGT in Tet1. They identified a 45 amino-acid sequence which is sufficient to mediate TET1-OGT interaction, as well as a single point mutation which abrogates the TET1-OGT interaction. They did this mostly using in vitro biochemical methods, along with some in vivo work to support it.

The mapping of the site is an interesting finding and the point mutant is a fine tool for further dissection of the contribution of the TET1-OGT interaction to regulating TET1 activity as well as genome maintenance. This paper hints at potentially interesting functions of the TET1-OGT interaction, with some data derived from mESCs and developing zebrafish embryos. However, the function of this interaction in regulating TET proteins, hmC levels, or downstream transcriptional dynamics at specific genomic loci or in different chromatin environments remains somewhat speculative, and further experiments are required in order to build a model of how the TET1-OGT may actually influence transcriptional dynamics.

Also basic questions remain unanswered, e.g. does the interaction per se activate TET1 or is it the post-translational modification of TET1 and which sites are affected?

Why do the authors jump from human HEK to mouse ESCs and do the functional assay in fish rescuing a Tet2/3 KO with TET1?

Throughout, rigorous controls, careful quantification and statistical analyses are missing.

Figure 1A-C:

The co-factor UDP-GlcNAc is certainly required for in vitro O-GlyNAcylation. However, it is unclear if OGT can bind to TET1 without this co-factor. In other words, is active o-GlyNAcylation required for this interaction?

Figure 1E-F:

The authors suggest a decrease in mC and hmC levels when OGT is overexpressed in mESCs. and attribute this is to stimulation of TET1 activity by OGT. However, overexpression of OGT (or many other nuclear proteins) often comes with pleiotropic effects. To rule out these possible side-effects, the authors should repeat these immunostainings and slot blots on cells where TDG is depleted, and look for an increase in accumulation of TET1 oxidation products hmC, caC and fC when OGT is overexpressed.

The immunofluorescence images are difficult to interpret – the images suggest massive changes in mC and hmC levels in cells, whereas the slot blots suggest very slight changes in levels of these modifications (these should be quantified: the AP stained slot blot for mC shows what appears to be less DNA loaded on the membrane for the OGT++ cells). mC staining in WT cells shows strong cytosolic background/unspecific staining, whereas this background is absent in OGT++ cells, which suggests problems with the experiment itself, or unusually altered display contrast settings.

Co-staining with DAPI would help identify nuclei and evaluate the changes in mC/hmC intensity in single cells and would help with basic quantification of the signal. Additionally, it is somewhat unlikely that all cells would show such a change in mC/hmC from transient overexpression of OGT – this would suggest 100% transfection efficiency, and 100% penetrance of the overexpression phenotype. These efficiencies might be possible, but needs to be shown in detail, possibly by co-expressing cytosolic GFP under the same overexpression promoter, or by tagging OGT itself with a fluorophore.

These issues, along with the fact that transfection efficiency in Figure 5 appears to be around 25%, point to problems with the IF experiments, and should be entirely repeated, with adequate controls and some degree of quantification.

Finally, no information is given on how these images were acquired, whether identical acquisition settings were used for control and OGT+ conditions, and how the contrast/offset was adjusted for visualization of these images. Additional images, perhaps with lower magnification showing more cells should be supplied in supplemental data.

Figure 3:

Here the authors identify a residue necessary for TET1-OGT interaction. Since this residue seems to be conserved in all three TET proteins, it would important to show whether this residue is minimally required for OGT interaction of each TET protein.

If that residue is required for OGT interacting with TET2 and TET3, then these would make far more interesting candidates for the rescue experiment described in Figure 6.

In Figure 3E, the D2018A mutant pulled-down OGT, which the authors suggest are high-molecular weight complexes, possibly involving TET2 and HCFC. It would be helpful to show that this is the case by probing for TET2 and HCFC. This would help with further dissection of OGT's direct interaction with TET vs. indirect interaction via TET-containing complexes.

Figure 5:

Here the authors show that TET1-D2018A is catalytically active when overexpressed in HEK cells. However, the difference in hmC levels is more visible here than in other slot blots, but the authors suggest that hmC levels are comparable, and that the mutation doesn't notably affect hmC generation, which seems misleading.

More importantly, however, the overexpression itself may mask subtle changes is the rate of hmC production. Therefore, given the availability of Cas9-engineered mESCs expressing the D2018A mutant, it would be more relevant to look at hmC levels in mESCs, and see how hmC levels change/are maintained when TET1-D2018A is expressed at more physiologically relevant levels.

To further characterize this interaction, a chromatin binding assay comparing soluble vs. chromatin-bound fractions of TET1 and OGT may give further insight into how TET1 recruits OGT to chromatin. Ideally, genome-wide methods such as RNA-seq and/or ChIP-seq would provide definitive evidence of localization and transcriptional regulation mediated by the TET1-OGT interaction, and would shed light on the chromatin/genomic environment where these interactions are active. These data would allow to build a more robust model of how TET1-OGT influences transcriptional dynamics, and where this interaction takes place.

It should be noted that the imaging data here in Figure 5 is much better interpretable than in Figure 1 – the images here display the transfection efficiency and how expression levels vary with transient transfection.

Figure 6:

Here the authors partially rescue the low-runx phenotype found in *tet2/3* double-mutant zebrafish larvae by overexpressing mouse TET1 and TET1-D2018A. Given the overlapping requirements of TET2 and TET3 as described in Li et al., 2015, it would be more relevant to rescue with TET2 or TET3 proteins in which OGT interaction was abrogated (as suggested above).

It is unclear how the embryos were sorted into "high" or "low" runx categories, and whether this was performed in a blind/randomized way. Additionally, for each experiment, the number of embryos and number of clutches should be shown.

In Figure 6C, the slot blot is not quantified and changes in hmC levels are quite difficult to perceive, thus the claim that hmC levels are affected with TET1 injection is a bit tenuous.

The claim that TET1-OGT promotes TET1 activity is a bit strong based on this experiment. While the point mutant may well affect TET1-OGT interaction, D2018A may also disrupt the binding of other proteins to TET1, as well as the stability of the protein itself.

Furthermore, the rescue of the high runx phenotype may be a distant downstream effect. Do the authors speculate that TET1-OGT directly modulates runx transcription? Or is this a downstream result from TET1-OGT somehow affecting Notch signalling?

---

## [Author Response]

Essential revisions:The reviewers and editors feel that the biological function of the interaction between OGT and TET1 in ESCs requires further investigation, as the presented phenotypes are complex and somewhat confusing. Specifically:1) ES cells with homozygous TET1 D2018A mutation have significantly elevated levels of TET2, as well as a higher proportion of TET2 that co-fractionates with OGT and HCF1 within high molecular weight complexes (Figure 6). The elevated levels of TET2 in TET1 D2018A mutant cells may reflect increased stability through formation of large complexes with OGT, HCF1, and potentially other proteins. Although TET2 function is not the primary focus of the study, the phenotype of TET1 D2018A mutant cells, in which 5mC is redistributed and seemingly reduced while 5hmC levels are unchanged, merits further examination. The authors should test whether the stability or expression of TET2 is increased in D2018A mutant cells.

We now include analysis of TET2 mRNA levels (RT-qPCR, RNA-seq), which show that there are no significant differences in levels of TET2 mRNA. To examine stability of TET2 protein we performed cycloheximide block experiments, which show no notable difference in half-life between WT and D2018A mESCs.

2) The images in Figure 7D suggest that ES cells with the TET1 D2018A mutation have reduced 5mC levels, but the immunofluorescence staining only gives limited information on how TET1-OGT interaction affects the distribution of 5mC and 5hmC. The authors conclude that the distribution of 5mC is altered, but the overall levels are not directly measured, and the interpretation that 5hmC is similar between mutant and WT, but that 5mC (the substrate) is redistributed and potentially reduced is confusing. Changes in bulk 5mC levels should be measured in these cells. The authors should also conduct genome-wide profiling of 5mC and 5hmC, and potentially of TET1, TET2, TET3 and OGT chromatin binding.

a) We assayed bulk levels of 5mC and 5hmC in WT and TET1 D2018A cells by mass spectrometry. These new data showed reduced 5mC levels in D2018A cells compared to WT and comparable levels of 5hmC in both cell types, consistent with the data in the previous submission.

b) We collaborated with Joe Ecker’s lab for bisulfite-seq and Chuan He’s lab for hmC-Seal analyses in WT and TET1 D2018A cells. These results support the conclusions from bulk studies of 5mC and 5hmC levels and suggest that 1) 5mC levels are lower in D2018A cells but 5mC distribution is largely unaltered in unique sequences, and 2) 5hmC levels are similar in the two cell types but 5hmC undergoes a redistribution at a small subset of gene bodies. These higher-resolution, genome-wide analyses replace the immunofluorescence data in the previous submission (old Figure 7D), which predominantly queried repetitive regions and showed 5mC is also decreased in these regions.

3) Although reduction of TET1 activity would not be predicted to cause reduction of 5mC levels on DNA, the authors suggest increased TET2 activity may compensate for reduced TET1 activity in D2018A ES cells. Direct measurement of the 5mC oxidation activities of TET1 and TET2 complexes in wild type and D2018A cells would be difficult, but the authors may be able to restore normal levels and distribution of 5mC in D2018A cells by overexpression of the GFP-C45 fusion, which binds OGT in vitro (Figure 3). This would provide evidence that the OGT interaction domain of TET1 has two important functions, activation of TET1 activity and titration of OGT to limit activation of TET2. Knockdowns of TET2 and/or TET3 in the WT and mutant cells could also isolate the effects of the TET1 mutation.

This question is a very interesting and important one and we thank the reviewers for raising it. However, rigorous analyses are beyond the scope of the current submission. Overexpression the GFP-C45 construct is problematic because it may affect multiple non-TET OGT targets by sequestering OGT. The second approach suggested was to knock down TET2 and/or TET3. Analyses suggest that the D2018A mutation changed abundance and/or composition of TET containing nuclear complexes. Therefore to meaningfully address the contribution of TET2/3 to the TET1 D2018A phenotype will require amounts in excess of what can be obtained with knockdown, since the effects of this perturbation on high molecular weight complexes will have to be queried. To obtain sufficient material for informative analyses, we initially focused on TET2, the increase in abundance or activity of which could underlie the reduced 5mC in D2018A mESCs. We attempted to knock out *Tet2* in the D2018A cell lines multiple times using multiple strategies, all without success. As a result, we believe that this genotype is lethal, and plan to employ the auxin-inducible degron system to turn over TET2 (and 3). Once appropriate cell lines are generated it will be possible to meaningfully assess the effects of perturbing other TETs in the D2018A background. The outcomes of these experiments will inform the complexity of the interaction between all three TETs and OGT in ESCs, but will not alter the conclusions of this manuscript.

4) The authors should add more analysis to the RNA-seq data. For example, how does the set of differentially expressed genes overlap with OGT ChIP-seq targets in published studies?

To facilitate comparison between RNA-seq, bisulfite-seq, and hmC-Seal we performed all three analyses on the same batches of cells. We compared the differentially expressed genes (DEGs) with differentially methylated regions (DMRs) and differentially hydroxymethylated regions (DhMRs). There is a correlation between genes containing DMRs and differentially expressed genes, suggesting that changes in CpG modifications may underlie some of the gene expression changes. We did not identify enough DhMRs by 5hmC-Seal to make statistically meaningful comparisons between DEGs and DhMRs. There was no significant overlap between DEGs and OGT occupancy. One caveat when comparing our –seq data to published data sets is that we use XX mESCs and published studies use XY mESCs. OGT is X-linked and more highly expressed in XX mESCs than XY mESCs because X-inactivation has not yet occurred in mESCs. The greater expression of OGT (and other X-linked genes) greatly complicates comparisons of XX and XY –seq based data sets.

[Editors’ note: the author responses to the first round of peer review follow.]

The manuscript has changed significantly since the first submission. We have made revisions and added new data in response to the reviewer comments, and we include a point-by-point response to the reviewers with this submission. The new data extends our analysis of the interaction between the epigenetic regulator TET1 and the nutrient sensor OGT both in vitroand ex vivo. We demonstrate that stimulation of TET1’s catalytic activity by OGT in vitro requires TET1 *O-*GlcNAcylation. We show that in mouse embryonic stem cells (mESCs) a point mutation that disrupts the direct OGT-TET1 protein-protein interaction alters the composition of TET-containing high-molecular-weight protein complexes, changes the genome-wide distribution of 5mC, and causes significant changes in gene expression. Since OGT mediates the post-translational modification of over 1,000 proteins in response to the cell’s nutrient status and TETs regulate gene expression, our work connects metabolism to epigenetics and suggests new avenues of investigation for both fields.

Reviewer #1:[...] Overall, the authors proposed and tested a novel hypothesis on the interaction between OGT and TET1. However, it is very confusing to see that the D2018A mutant of TET1 lost the interaction with OGT but still retained enzymatic activity in 293T cells. A careful and through study is needed to justify the major conclusions in this manuscript.1) It is unclear if OGT is required for the enzymatic activity of TET1 in mammalian cells. The authors showed that the D2018A mutant was still able to convert 5mC to 5hmC in 293T cells. The enzymatic activity of the D2018 mutant was similar to that of wild type TET1 (Figure 5B and C). As the expression of OGT is very high in 293T cells, it indicates that without the interaction with OGT, TET1 is still able to oxidize 5mC, which contradicts with the major conclusions of the manuscript.

We apologize for our lack of clarity. We did not intend to convey that these data showed OGT is required for TET1 activity. Our data show that OGT stimulates activity of the TET1 catalytic domain in vitro. The purpose of expressing the D2018A TET1 mutant in HEK293T cells was to query whether the mutation eliminated TET1 activity in the context of the entire protein, not to assess whether the OGT-TET1 interaction affected TET1 activity in cells. OGT levels are very low in our HEK293T cells, so to address whether OGT is affecting TET1 activity it would be necessary to co-transfect in OGT with TET1. We did not attempt this experiment because of the difficulty in quantitating the expression of OGT, TET1, and 5hmC in individual cells in a transient transfection assay.

2) The authors stated "we mutated these residues individually and found that D2018A eliminated detectable interaction between FLAG-tagged TET1 CD and myc-OGT". However, in the Figure 3D, only D2018A had been examined. It is unclear if other mutations abolish the interaction between TET1 and OGT. In particular, D2018 is not mutated in TET1 CD mt2. But TET1 CDmt2 still lost the interaction with OGT, indicating that other residues such as V2021 and T2022 may play more important roles for the interaction.

Reviewer 1 is absolutely correct in that residues in addition to D2018 may be necessary for TET1 to interact with OGT, and we have included this point in the revised submission. Regardless of whether additional point mutations alter the interaction, the D2018A mutation gives us the tools to query the biological role of the OGT-TET1 interaction.

3) The authors have not demonstrated if the enzymatic activity of OGT is required for TET1-dependent functions. The authors need to show if the enzymatic dead OGT is able to activate TET1 both in vitro and in vivo.

We have added new data showing that *O*-GlcNAcylation of the TET1 catalytic domain by OGT is necessary for stimulation of TET1 activity in vitro. Given these in vitro results, we did not pursue the effect of expressing catalytically dead OGT on TET1 activity in cells.

4) The study on the TET1 mutant is incomplete. It is unclear if OGT is able to promote the enzymatic activity of the mutant TET1 in vitro.

Our new data (Figure 4) show that OGT does not promote enzymatic activity of mutant TET1 catalytic domain in vitro.

5) The authors showed a conserved interacting motif existed in both TET2 and TET3. Does TET2 or TET3 interact with OGT via the same motif? Does OGT promote TET2 and TET3 as well?

It would certainly be interesting to determine whether the conserved interacting motif is also necessary and sufficient for interaction of TET2 and TET3 with OGT and whether OGT stimulates other TETs. These experiments are underway and are beyond the scope of this manuscript.

6) It is very strange to see that all the TPR motifs of OGT are important for the interaction. Usually, different TPR motifs interact with different OGT substrates. If all the TPR repeats are required for the interaction with TET1, will other OGT substrates, such as HIF-1, affect the interaction between OGT and TET1? Especially HIF-1 is in the same complex with TET1 and OGT. Is the catalytic domain of OGT also required for the interaction with TET1?

TETs are unusual among OGT targets in that they stably interact with OGT. For most other targets the interactions are transient and not detectable by co-IP. We could suggest that the interaction with multiple TPRs may be one of the reasons that underlie this unusually strong interaction between OGT and one of its targets. In addition, as this reviewer correctly points out, additional interaction interfaces mediated by other proteins that co-IP with OGT and TET1, such as HCF-1, may be affecting the stability of the OGT-TET1 interaction in ESCs. As a result we elected not to speculate on the significance of the requirement for multiple TPRs for the OGT-TET1 interaction.

Reviewer #2:Hrit et al. follow up on work from several groups showing O-GlcNAc transferase (OGT) binds to TET proteins, and find that O-GlcNAc modification of TET1 stimulates its methylcytosine oxidation activity in vitro. […] However, the extent to which the D2018A mutant prevents O-GlcNAc modification and stimulation of TET1 activity by OGT is not thoroughly tested. In addition, the study would be strengthened by a deeper analysis of how TET1's activity is regulated by O-GlcNAc modification (see below).1) Although there is a non-quantitative hmC blot that shows reduced hmC in fish rescued with mutant TET1 mRNA (Figure 6C), it is not clear whether reduction of meC oxidation is due to reduced O-GlcNAc modification, reduced TET1/OGT physical interaction, or both. The relative contributions of TET1/OGT interaction and O-GlcNAc modification on TET1 activity could be measured in the authors' in vitro system.

As requested, we queried the roles of the *O*-GlcNAc modification and of the protein-protein interaction in TET1 stimulation, using the TET1 D2018A mutant catalytic domain. The revised submission includes new data showing that *O-*GlcNAcylation of the TET1 catalytic domain by OGT is necessary for stimulation of TET1 activity in vitro.

2) Perhaps the most interesting new finding is the demonstration that O-GlcNAc modification of TET1 by OGT strongly stimulated its activity (Figure 1B and C). This observation would be strengthened by examination of how O-GlcNAc modification leads to enhanced catalytic activity. Does this modification increase the affinity of TET1 for meC? Does O-GlcNAc affect the Km or Vmax of the enzyme? These questions should be testable with existing reagents used in Figure 1.

We agree that gaining more mechanistic insight into how *O*-GlcNAcylation stimulates TET1 activity will be very interesting. These experiments are complex because there are at least 8 *O*-GlcNAcylation sites on the TET1 catalytic domain. We are currently developing the reagents (*O*-GlcNAc site mutants) to meaningfully query the mechanisms (which may be multiple) by which OGT stimulates TET1.

Reviewer #3:[…] This paper hints at potentially interesting functions of the TET1-OGT interaction, with some data derived from mESCs and developing zebrafish embryos. However, the function of this interaction in regulating TET proteins, hmC levels, or downstream transcriptional dynamics at specific genomic loci or in different chromatin environments remains somewhat speculative, and further experiments are required in order to build a model of how the TET1-OGT may actually influence transcriptional dynamics.Also basic questions remain unanswered, e.g. does the interaction per se activate TET1 or is it the post-translational modification of TET1 and which sites are affected?

In the revised submission we now include new data showing that it is the posttranslational modification of TET1 that is responsible for the bulk of the stimulation of its activity. Determining which sites are the relevant ones for stimulation is beyond the scope of this work.

Why do the authors jump from human HEK to mouse ESCs and do the functional assay in fish rescuing a TET2/3 KO with TET1?

We employed different systems to ask questions that were most effectively answered using that system. For example, to query the role of the OGT-TET1 interaction in vivo, we employed zebrafish. The HEK239T cells express very little TET1 (or any TET) and very little OGT, which combined with the efficiency of transfection into these cells, facilitated rapid analysis of interaction domains ex vivo. Mouse ESCs express abundant TET1 and OGT, and introduction of the D2018A mutation in these cells allows analysis of the effects of the mutation on cytosine modifications and gene expression.

Throughout, rigorous controls, careful quantification and statistical analyses are missing.Figure 1A-C:

*The co-factor UDP-GlcNAc is certainly required for* in vitro *O-GlyNAcylation. However, it is unclear if OGT can bind to TET1 without this co-factor. In other words, is active o-GlyNAcylation required for this interaction?*

OGT and TET1 interact without UDP-GlcNAc, as shown in Figure 3.

Fig1E-F:The authors suggest a decrease in mC and hmC levels when OGT is overexpressed in mESCs. and attribute this is to stimulation of TET1 activity by OGT. However, overexpression of OGT (or many other nuclear proteins) often comes with pleiotropic effects. To rule out these possible side-effects, the authors should repeat these immunostainings and slot blots on cells where TDG is depleted, and look for an increase in accumulation of TET1 oxidation products hmC, caC and fC when OGT is overexpressed.

Reviewer 3 correctly points out that analysis of OGT overexpressing cells is complicated by both the turnover of 5fC and 5caC by TDG and by pleiotropic effects of OGT over expression itself. Rather than perform the suggested TDG depletion experiments, we have removed the data. Instead, we present analysis of cytosine modifications in ESCs bearing the D2018A mutation perturbing the OGT-TET1 interaction (Figure 6-7). Because this manipulation doesn’t affect OGT levels, it is less likely to have pleiotropic effects and is therefore more informative.

The immunofluorescence images are difficult to interpret – the images suggest massive changes in mC and hmC levels in cells, whereas the slot blots suggest very slight changes in levels of these modifications (these should be quantified: the AP stained slot blot for mC shows what appears to be less DNA loaded on the membrane for the OGT++ cells). mC staining in WT cells shows strong cytosolic background/unspecific staining, whereas this background is absent in OGT++ cells, which suggests problems with the experiment itself, or unusually altered display contrast settings.

While the slot blots and immunostaining images in Figure 1 have been removed, we have additional blots and immunostaining images in the revised submission. For more quantitative analysis, relevant slot blots are reproduced 3+ times, results normalized to input, and expressed graphically with a representative slot blot shown. Immunostaining can sometimes give background, but this background does not invariably mean that contrast settings have been altered. To make it clear that all images are treated the same way, we include a description of image capture and analysis in the revised Materials and methods.

Co-staining with DAPI would help identify nuclei and evaluate the changes in mC/hmC intensity in single cells and would help with basic quantification of the signal. Additionally, it is somewhat unlikely that all cells would show such a change in mC/hmC from transient overexpression of OGT – this would suggest 100% transfection efficiency, and 100% penetrance of the overexpression phenotype. These efficiencies might be possible, but needs to be shown in detail, possibly by co-expressing cytosolic GFP under the same overexpression promoter, or by tagging OGT itself with a fluorophore.

A stable cell line over expressing OGT, not transient transfection, was employed in Figure 1 of the first submission, so it is possible that all cells showed a change in 5mc/5hmC. The suggestion of including the DAPI staining is helpful, and all immunostainings in the revised figure include a DAPI panel.

These issues, along with the fact that transfection efficiency in Figure 5 appears to be around 25%, point to problems with the IF experiments, and should be entirely repeated, with adequate controls and some degree of quantification.

The concerns with IF experiments in Figures 1 and 5 were about transfection efficiency, which in turn brought into question the quantitation. Both these IF experiments have been removed and replaced by analysis of ESC lines bearing the TET1 D2018A mutation, which eliminates concerns about transfection efficiency and the accompanying quantitation.

Finally, no information is given on how these images were acquired, whether identical acquisition settings were used for control and OGT+ conditions, and how the contrast/offset was adjusted for visualization of these images. Additional images, perhaps with lower magnification showing more cells should be supplied in supplemental data.

As stated above, we include a description of image capture and analysis in the revised Materials and methods. Scale bars are now also included.

Figure 3:Here the authors identify a residue necessary for TET1-OGT interaction. Since this residue seems to be conserved in all three TET proteins, it would important to show whether this residue is minimally required for OGT interaction of each TET protein.If that residue is required for OGT interacting with TET2 and TET3, then these would make far more interesting candidates for the rescue experiment described in Figure 6.

While it would indeed be interesting to map the requirements for OGT interaction in each of the three TETs, this endeavor is beyond the scope of the current work.

In Figure 3E, the D2018A mutant pulled-down OGT, which the authors suggest are high-molecular weight complexes, possibly involving TET2 and HCFC. It would be helpful to show that this is the case by probing for TET2 and HCFC. This would help with further dissection of OGT's direct interaction with TET vs. indirect interaction via TET-containing complexes.

Reviewer 3 echos our discussion point that the interaction between OGT and TET1 D2018A in ESCs may reflect additional contacts between TET1 and other proteins in the TET-OGT-HCF high molecular weight complexes. To address this point we analyzed high molecular weight complexes in TET1 wild type and D2018A mutant ESCs. These data (Figure 6 in the revised submission) show that the D2018A mutation has effects on all three TET-containing high molecular weight complexes.

Figure 5:Here the authors show that TET1-D2018A is catalytically active when overexpressed in HEK cells. However, the difference in hmC levels is more visible here than in other slot blots, but the authors suggest that hmC levels are comparable, and that the mutation doesn't notably affect hmC generation, which seems misleading.More importantly, however, the overexpression itself may mask subtle changes is the rate of hmC production. Therefore, given the availability of Cas9-engineered mESCs expressing the D2018A mutant, it would be more relevant to look at hmC levels in mESCs, and see how hmC levels change/are maintained when TET1-D2018A is expressed at more physiologically relevant levels.

We have performed the analysis of 5mC and 5hmC levels/distribution in D2018A ESCs, as requested. They are included as Figure 7 in the revised submission.

To further characterize this interaction, a chromatin binding assay comparing soluble vs. chromatin-bound fractions of TET1 and OGT may give further insight into how TET1 recruits OGT to chromatin. Ideally, genome-wide methods such as RNA-seq and/or ChIP-seq would provide definitive evidence of localization and transcriptional regulation mediated by the TET1-OGT interaction, and would shed light on the chromatin/genomic environment where these interactions are active. These data would allow to build a more robust model of how TET1-OGT influences transcriptional dynamics, and where this interaction takes place.

We have performed the RNA-seq analysis, as requested. The new data are included in Figure 7 of the revision.

It should be noted that the imaging data here in Figure 5 is much better interpretable than in Figure 1 – the images here display the transfection efficiency and how expression levels vary with transient transfection.Figure 6:Here the authors partially rescue the low-runx phenotype found in tet2/3 double-mutant zebrafish larvae by overexpressing mouse TET1 and TET1-D2018A. Given the overlapping requirements of TET2 and TET3 as described in Li et al., 2015, it would be more relevant to rescue with TET2 or TET3 proteins in which OGT interaction was abrogated (as suggested above).

It would be interesting to determine whether the core finding, that the interaction of a TET with OGT is necessary for normal TET-mediated rescue of a developmental phenotype in zebrafish, holds to all three TETs. However, that line of investigation is beyond the scope of this work.

It is unclear how the embryos were sorted into "high" or "low" runx categories, and whether this was performed in a blind/randomized way. Additionally, for each experiment, the number of embryos and number of clutches should be shown.

In an effort to improve clarity we now include a supplementary figure that includes representative examples of high and low *runx1* categories (Supplementary file 3). This figure also includes the number of embryos for each experimental condition for each of the 5 clutches examined. We make clear in the Materials and methods that analysis of high vs. low *runx1* expression was performed without prior knowledge of experimental condition (blind) and that embryos were genotyped after in situ analysis.

In Figure 6C, the slot blot is not quantified and changes in hmC levels are quite difficult to perceive, thus the claim that hmC levels are affected with TET1 injection is a bit tenuous.

We now include a graph quantifying changes in hmC across three dot blots and provide a detailed explanation of our quantification strategy in the Materials and methods. This analysis demonstrates that rescue with TET1 leads to a small but statistically significant increase in 5hmC whereas injections of the mutant version of TET1 do not.

The claim that TET1-OGT promotes TET1 activity is a bit strong based on this experiment. While the point mutant may well affect TET1-OGT interaction, D2018A may also disrupt the binding of other proteins to TET1, as well as the stability of the protein itself.

The reviewer makes an important point. We have adjusted the results summary of the zebrafish data to more clearly indicate that the effects of D2018A mutation in zebrafish may be direct or indirect.

Furthermore, the rescue of the high runx phenotype may be a distant downstream effect. Do the authors speculate that TET1-OGT directly modulates runx transcription? Or is this a downstream result from TET1-OGT somehow affecting Notch signalling?

Understanding the basis of the rescue, whether it is through a direct or indirect effect, is interesting but the basis of another manuscript.